# High-performance light-driven heterogeneous CO$_2$ catalysis with near-unity selectivity on metal phosphides

Yang-Fan Xu[1], Paul N. Duchesne[1], Lu Wang[1], Alexandra Tavasoli[1], Feysal M. Ali[1], Meikun Xia[1], Jin-Feng Liao[2], Dai-Bin Kuang [2✉] & Geoffrey A. Ozin [1✉]

Akin to single-site homogeneous catalysis, a long sought-after goal is to achieve reaction site precision in heterogeneous catalysis for chemical control over patterns of activity, selectivity and stability. Herein, we report on metal phosphides as a class of material capable of realizing these attributes and unlock their potential in solar-driven CO$_2$ hydrogenation. Selected as an archetype, Ni$_{12}$P$_5$ affords a structure based upon highly dispersed nickel nanoclusters integrated into a phosphorus lattice that harvest light intensely across the entire solar spectral range. Motivated by its panchromatic absorption and unique linearly bonded nickel-carbonyl-dominated reaction route, Ni$_{12}$P$_5$ is found to be a photothermal catalyst for the reverse water gas shift reaction, offering a CO production rate of 960 ± 12 mmol g$_{cat}^{-1}$ h$^{-1}$, near 100% selectivity and long-term stability. Successful extension of this idea to Co$_2$P analogs implies that metal phosphide materials are poised as a universal platform for high-rate and highly selective photothermal CO$_2$ catalysis.

[1] Materials Chemistry and Nanochemistry Research Group, Solar Fuels Cluster, Department of Chemistry, University of Toronto, Toronto, Ontario M5S 3H6, Canada. [2] MOE Key Laboratory of Bioinorganic and Synthetic Chemistry, Lehn Institute of Functional Materials, School of Chemistry, Sun Yat-sen University, 510275 Guangzhou, Guangdong, P. R. China. ✉email: kuangdb@mail.sysu.edu.cn; g.ozin@utoronto.ca

In the field of heterogeneous catalysis, gaseous or liquid reactants undergo chemical reactions on the surface of a solid material. These reactions can be enabled thermochemically, electrochemically, biochemically or photochemically[1–8]. The surface reactivity of catalytically active sites is typically discussed in terms of electronic, geometric and support effects[9–15]. Moreover, as the number of catalytically active sites scales with the surface area of the catalyst, size matters, especially at the nanoscale. Yet most heterogeneous catalysts exhibit surface structure heterogeneity that depend upon particle size and exposed facets, which influences reactivity and selectivity patterns. Furthermore, the size distribution and facet exposure can change over the course of a catalytic reaction, having a deleterious effect on selectivity patterns and long-term stability. Ideally, a heterogeneous catalyst comprised of single-size, high surface area nanoparticles, with uniform shape and facet exposure that remains stable under reaction conditions could overcome these problems; in practice however, they have rarely been achieved.

Metal phosphides form a class of solids that provide the envisioned atomic and crystalline perfection with a structure based upon highly dispersed metal nanoclusters chemically integrated in a P lattice. These materials abound with diverse stoichiometry across a wide range of metals in the periodic table offering opportunities as catalysts for electrochemical hydrogen evolution reactions, hydro-processing, and so forth[16–20]. However, their appealing potential for heterogeneous (photo)catalytic $CO_2$ hydrogenation remains to be unlocked.

Herein, we focus attention on an archetypal nickel phosphide, $Ni_{12}P_5$, with a unique surface structure based upon well-separated few-atom Ni nanoclusters. This structure allows $Ni_{12}P_5$ to function as an exceptionally active, selective, and stable heterogeneous catalyst for the photothermal reverse water gas shift (RWGS) reaction under light stimulation. Moreover, this concept is further extended to the construction of $Co_2P$ analogs, thus revealing a bright future for the earth-abundant transition metal phosphides as a class of heterogeneous catalysts for $CO_2$ hydrogenation application. This potential is demonstrated by the high conversion rates and selectivity and long-term stability observed for these transition metal phosphides in heterogeneous $CO_2$ hydrogenation reactions driven by solar energy.

## Results

**Material synthesis and characterizations**. Nickel phosphide materials were prepared by reducing nickel phosphate oxides under $H_2$ flow. The XRD patterns, presented in Fig. 1a, showed that, in accordance with the Ni/P ratio of the chemical precursors, the as-prepared materials formed in the tetragonal $Ni_{12}P_5$ phase. In addition, a series of $SiO_2$-supported $Ni_{12}P_5$ samples with different loadings (denoted by x wt% $Ni_{12}P_5$/$SiO_2$, where the x is the weight percent of $Ni_{12}P_5$ determined by ICP-OES) were prepared via wet impregnation, with XRD patterns indicating that the composition and phase remained unchanged. The as-synthesized $N_{12}P_5$ exhibited a black color (Supplementary Fig. 1), and so diffuse reflectance UV-vis-NIR spectra were recorded to quantify its light absorption properties. As shown in Fig. 1b, the $Ni_{12}P_5$ nanoparticles exhibited broadband absorption throughout UV-vis-NIR region. Loading the $Ni_{12}P_5$ on $SiO_2$, however, resulted in reduced optical absorption due to the dilution by the $SiO_2$ support.

The morphology and crystalline structure of the samples were subsequently characterized by transmission electron microscopy (TEM). The low-magnification TEM image (Supplementary Fig. 2b) and the corresponding particle size distribution statistics (Supplementary Fig. 3a) showed that the diameters of unsupported $Ni_{12}P_5$ nanoparticles had a wide distribution with a mean

diameter of 86 nm. Distinctive lattice fringe spacings of 0.604 nm and 0.401 nm, with an angle of 69°, were observed in the corresponding high-resolution TEM image (Fig. 1c), which is in accordance with the (110) and (101) planes of tetragonal $Ni_{12}P_5$. From the TEM images of $Ni_{12}P_5$/$SiO_2$ samples (Supplementary Fig. 2c–e), mean particle diameters were distinctly smaller as the loading amount decreased, shrinking from about 13 nm in 10.4 wt% $Ni_{12}P_5$/$SiO_2$ to about 8 nm in 3.1 wt% $Ni_{12}P_5$/$SiO_2$ (Table 1). However, the high crystallinity was maintained, as shown by the HRTEM imaging on 10.4 wt% $Ni_{12}P_5$/$SiO_2$ sample (Fig. 1d), and its tetragonal phase was again confirmed by the observation of lattice spacings corresponding to the (112) and (330) planes.

X-ray photoelectron spectroscopy (XPS) was further performed to characterize the chemical states of the $Ni_{12}P_5$ surface. As shown in Supplementary Fig. 4a, the high-resolution Ni 2p region of the pristine $Ni_{12}P_5$ sample, before the photocatalytic reaction, consisted of two characteristic peaks located at 852.6 eV and 869.6 eV, which could be indexed to near zero-valent nickel. For the P 2p region (Supplementary Fig. 4b), two peaks were observed at 129.5 eV and 130.3 eV, and could be assigned to near zero-valent phosphorus. These values are in accordance with those reported in the literature[21,22]. Moreover, it is worth noting that the binding energy of Ni $2p_{3/2}$ is slightly higher than that of metallic Ni, while the P $2p_{3/2}$ binding energy is correspondingly lower than that of elemental P, thus indicating that Ni and P were partially positively and negatively charged, respectively, in $Ni_{12}P_5$.

**X-ray absorption spectroscopy measurements**. To further explore the structural features of $Ni_{12}P_5$, X-ray absorption spectroscopy measurements were conducted. As shown in the Ni K-edge X-ray absorption near-edge structure (XANES) spectra (Fig. 2a), the position and low intensity of the white line reveal the largely metallic character of $Ni_{12}P_5$, especially relative to the shifted and very intense peak observed for the strongly ionic NiO reference. These results thus provide very strong evidence supporting the low oxidation state of Ni atoms in $Ni_{12}P_5$. The bonding environment in $Ni_{12}P_5$ was further investigated using extended X-ray absorption fine structure (EXAFS) analysis. As shown in Fig. 2b, two coordination shells centered at 0.163 nm and 0.264 nm were observed in the Fourier-transformed Ni K-edge EXAFS spectra of $Ni_{12}P_5$, which resulted from Ni−P and Ni−Ni scattering paths, respectively. The dramatically reduced Ni–Ni bond peak in $Ni_{12}P_5$ relative to that of the Ni foil implied that the number of direct Ni–Ni bonds was reduced, due to the preferred formation of Ni–P bonds. Further curve-fitting results (Supplementary Table 1) indicated that Ni–Ni bond length was stretched from 2.480 Å in Ni foil to 2.510 Å in $Ni_{12}P_5$, and that the coordination number was significantly decreased from 12 in bulk Ni to 7 ± 1 in $Ni_{12}P_5$, which is in accordance to its theoretical crystal structure[23]. Thus, these EXAFS fitting results strongly support the formation of well-dispersed, low-coordinate Ni nanoclusters with slight structural distortions relative to the bulk metal.

**Photocatalytic performance**. Testing of photocatalytic $CO_2$ hydrogenation by $Ni_{12}P_5$ catalysts was performed in a batch reactor, with samples loaded onto a borosilicate glass microfiber filter. The reaction was conducted under an initial total pressure of 18 psi (about 1.2 bar, $CO_2/H_2 = 5:1$), with a light intensity of 2.3 W cm$^{-2}$ provided by an unfiltered Xe lamp. No external heating was adopted when using the batch reactor. Following completion of the photocatalytic reaction, products were quantified via gas chromatography. According to the GC traces (Supplementary Fig. 5), the product contained CO and a tiny

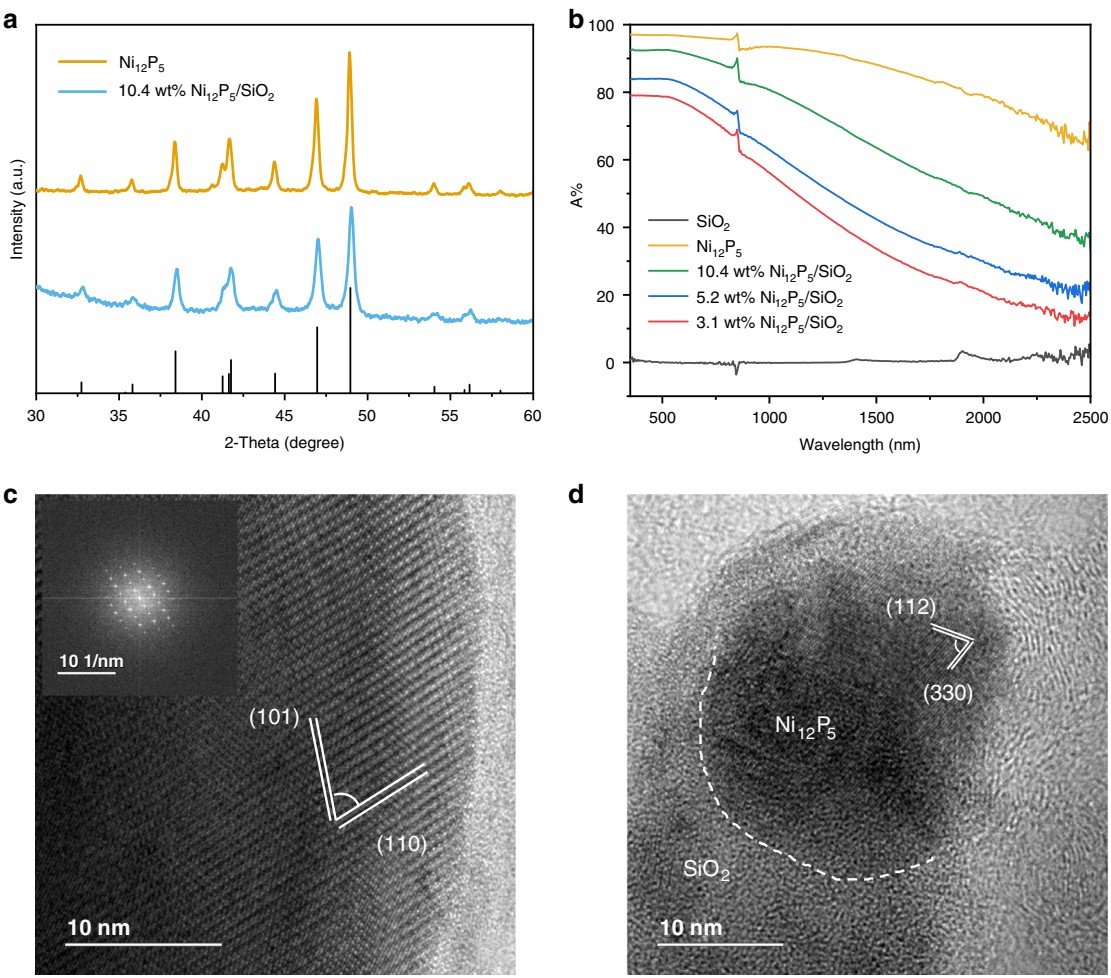

**Fig. 1 Characterization of Ni$_{12}$P$_5$ nanoparticles and the 10.4 wt% Ni$_{12}$P$_5$/SiO$_2$ nanocomposite. a** XRD patterns, black lines indicate the standard peaks of tetragonal Ni$_{12}$P$_5$ (PDF#74-1381). **b** UV-vis-NIR spectra of Ni$_{12}$P$_5$, SiO$_2$ and Ni$_{12}$P$_5$/SiO$_2$ samples loaded onto a binder-free borosilicate glass microfiber filter support (0.5 mg cm$^{-2}$). **c** HRTEM image of Ni$_{12}$P$_5$, inset is the corresponding FFT electron diffraction pattern. **d** HRTEM image of 10.4 wt% Ni$_{12}$P$_5$/SiO$_2$ composite.

**Table 1 Summary of properties and catalytic performances of representative Ni$_{12}$P$_5$ samples.**

| Sample | Ni$_{12}$P$_5$ TEM mean particle size (nm) | CO rate (mmol g$_{cat}^{-1}$ h$^{-1}$) | CO selectivity (%) |
|---|---|---|---|
| Ni$_{12}$P$_5$ | 86 ± 30 | 156 ± 3 | 99.5 ± 0.1 |
| 10.4 wt% Ni$_{12}$P$_5$/SiO$_2$ | 13 ± 7 | 960 ± 12 | 99.7 ± 0.1 |
| 5.2 wt% Ni$_{12}$P$_5$/SiO$_2$ | 9 ± 3 | 678 ± 13 | 99.7 ± 0.1 |
| 3.1 wt% Ni$_{12}$P$_5$/SiO$_2$ | 8 ± 4 | 334 ± 10 | 99.7 ± 0.1 |

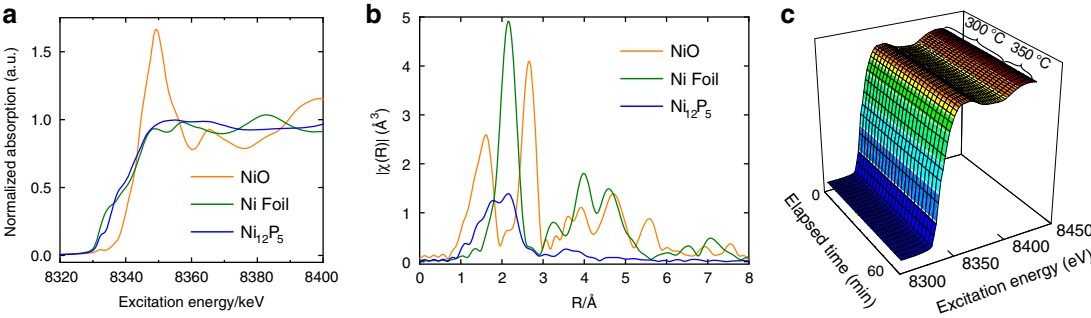

**Fig. 2 XAS measurements. a** Ni K-edge X-ray absorption near-edge structure (XANES) spectra, with Ni Foil and NiO powder included as references. **b** R-space spectra. **c** Time-resolved in situ XANES curves plotted in the simulated reaction condition (CO$_2$:H$_2$ = 5:1 gas flow, with a temperature of 300 °C or 350 °C).

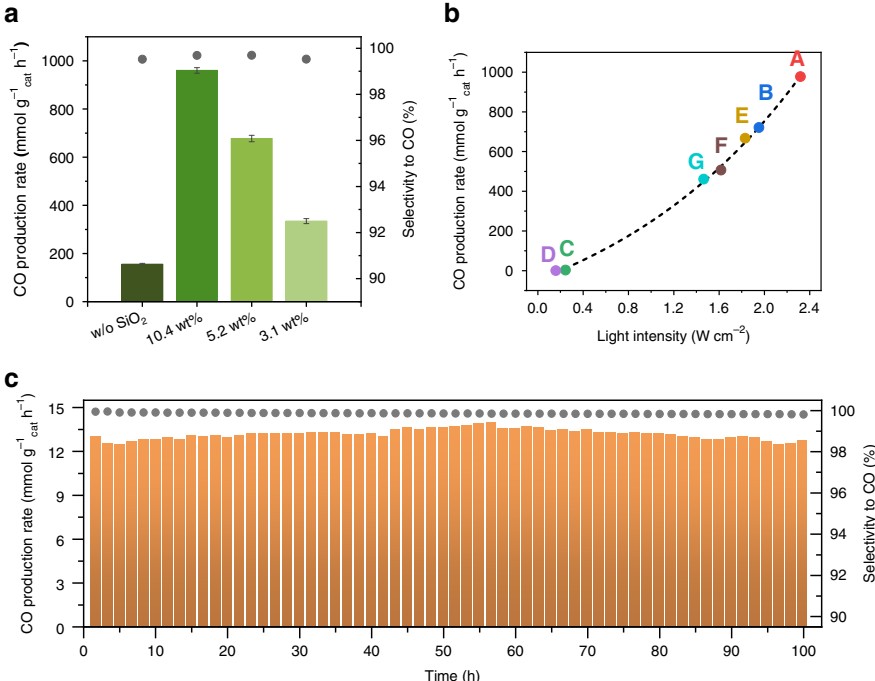

**Fig. 3 Light-driven RWGS reaction testing over Ni$_{12}$P$_5$ and Ni$_{12}$P$_5$/SiO$_2$. a** CO production rate and selectivity as a function of catalyst loading, tested in a batch reactor under 2.3 W cm$^{-2}$ illumination without external heating, the initial reactant comprise 15 psi of CO$_2$ and 3 psi of H$_2$ (total pressure of 1.2 bar). Error bars are based on standard division. **b** CO production rates plotted as a function of light intensity as controlled by (A) full-spectrum light without any filter, (B) a neutral-density (ND) 0.1 filter, (C) a ND 0.3 filter, (D) a ND 0.5 filter, (E) a 420 nm cut-off filter, (F) a 495 nm cut-off filter and (G) a 590 nm cut-off filter, tested in a batch reactor. **c** Long-term stability testing over 10.4 wt% Ni$_{12}$P$_5$/SiO$_2$ in a flow reactor, under both 0.8 W cm$^{-2}$ light illumination and thermal activation, with 2.5 sccm of CO$_2$ and 0.5 sccm of H$_2$ gas flow.

amount of CH$_4$, other products such as paraffin/olefin and methanol were not observed.

As summarized in Table 1 and plotted in Fig. 3a, a selectivity of 99.5% towards the RWGS route was observed, with an average CO production rate of 156 ± 3 mmol g$_{cat}$$^{-1}$ h$^{-1}$ being attained over unsupported Ni$_{12}$P$_5$ nanoparticles. Loading the Ni$_{12}$P$_5$ onto SiO$_2$ further accelerated the photocatalytic reaction rate, and a maximum rate of 960 ± 12 mmol g$_{cat}$$^{-1}$ h$^{-1}$ was achieved by the optimized 10.4 wt% Ni$_{12}$P$_5$/SiO$_2$ sample. This corresponds to a 5.2-fold enhancement that is quite reasonable, given that the correspondingly smaller particle size would enlarge the specific surface area and expose more accessible reaction sites per gram of catalyst, as observed from the Ni dispersion results (Supplementary Note 1). However, further lowering the loading amount results in a decrease of the CO production rate along with the turnover frequency value (for calculation details see Supplementary Note 1). One plausible explanation is the inferior light harvesting ability on lower loading samples (Fig. 1b).

As summarized in Supplementary Table 2, this rate of CO$_2$ hydrogenation to CO is a strong competitor to other reported photothermal catalysts, such as the commercial iron-chrome based catalyst, which delivered a CO production rate of 63.0 mmol g$_{cat}$$^{-1}$ h$^{-1}$ under the same test conditions. It should also be emphasized that selectivity toward the RWGS reaction was unchanged for the SiO$_2$-supported Ni$_{12}$P$_5$ samples, remaining at around 99.7%. Subsequent photocatalyst recyclability tests indicate that both the yield and selectivity were well retained over eight sequential runs (Supplementary Fig. 6), and XRD patterns from the spent sample confirmed that no Ni metal emerged following photocatalytic testing (Supplementary Fig. 7). XPS plots of the spent Ni$_{12}$P$_5$ sample following photocatalytic testing were also recorded and showcased identical binding energies for both Ni 2p and P 2p, confirming that the oxidation

state of both Ni and P elements remained unchanged at the Ni$_{12}$P$_5$ surface. To more closely monitor catalyst stability under simulated RWGS reaction conditions, time-resolved in situ XANES tests were performed under a 1:5 (v/v) flow of H$_2$/CO$_2$ gas at 300 °C over a 40 min period. Again, no significant changes were observed in the time-resolved XANES spectra (Fig. 2c), even after further increasing the temperature to 350 °C for another 20 min, thus confirming the high stability of Ni$_{12}$P$_5$ in the RWGS reaction.

Isotope tracing experiments were further conducted by replacing the normal $^{12}$CO$_2$ reagent gas with isotopically labeled $^{13}$CO$_2$ and using gas chromatography–mass spectrometry (GC–MS) to analyze the chemical products. As shown in Supplementary Fig. 8, the signal corresponding to $^{13}$CO ($m/z =$ 29) in the mass spectrum clearly confirmed that the product CO originated from the CO$_2$ feedstock rather than any carbon contaminants.

To elucidate the role of light in the catalytic performance of Ni$_{12}$P$_5$, a series of photocatalytic tests was first conducted under various light intensities by employing a set of neutral-density filters and high-pass cut-off filters. As plotted in Fig. 3b, both visible and NIR light were able to activate the Ni$_{12}$P$_5$ catalyst and promote the RWGS reaction. The plot of CO production rate versus light intensity was well fitted by an exponential function, indicating a predominantly photothermal mechanism, where the catalyst absorbs and convert the incident photon flux into heat to drive the subsequent catalytic reactions[24]. This is distinct to the photochemical process in which the chemical reaction is driven directly by the photo-generated charge carriers, and the reaction rate would usually scale with the light intensity. Furthermore, as shown in Supplementary Note 2, the internal quantum yield (IQY) was estimated, indicating a trend that lower Ni$_{12}$P$_5$ loading samples delivered inferior IQY values. As mentioned, this could

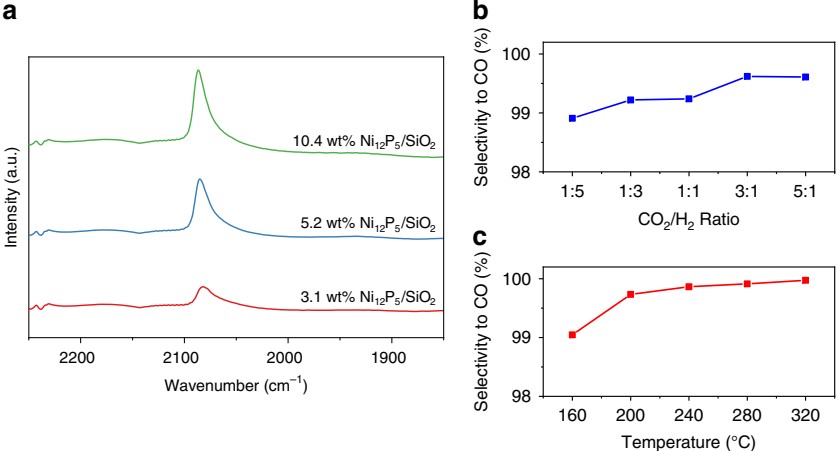

**Fig. 4 Determination of reaction pathway. a** In situ DRIFTS spectra of $Ni_{12}P_5/SiO_2$ samples with different loading amount. **b** The selectivity to the RWGS reaction over 10.4 wt% $Ni_{12}P_5/SiO_2$ sample, plots with respect to the $CO_2/H_2$ initial ratio, tested in a batch reactor under 2.3 W cm$^{-2}$ illumination without external heating. The initial total pressure was controlled as 18 psi (1.2 bar). **c** plots of the selectivity to the RWGS reaction over 10.4 wt% $Ni_{12}P_5/SiO_2$ sample with respect to the temperature, tested in a flow reactor under 0.8 W cm$^{-2}$ illumination and a gas flow of $CO_2$ (2.5 sccm) and $H_2$ (0.5 sccm).

relate to the light harvesting ability as better light absorption would contribute to a higher local temperature under light stimulation, and thereof a superior TOF value. Nevertheless, the IQYs of $Ni_{12}P_5$ and 10.4 wt% $Ni_{12}P_5/SiO_2$ are quite close. Considering only about one tenth of the active $Ni_{12}P_5$ has been utilized in the 10.4 wt% $Ni_{12}P_5/SiO_2$ sample when compared to pristine $Ni_{12}P_5$, it is conceivable that the former would be more competitive overall, especially if the system was to be scaled-up. The photothermal effect on 10.4 wt% $Ni_{12}P_5/SiO_2$ sample was further quantified by using ASPEN Plus software to estimate the local temperature of the catalyst surface based on the experimentally measured $CO_2$ conversion at reaction equilibrium[4]. Under 2.3 W cm$^{-2}$ irradiation and using a 5:1 ratio of $CO_2/H_2$, the photocatalytic reaction finally achieved 7.48% $CO_2$ conversion in the batch reactor after 11.5-h test (Supplementary Fig. 9). The corresponding predicted local temperature approached 401 °C on a wet basis and 381 °C on dry basis (Supplementary Table 3), thereby demonstrating the exceptional photothermal capabilities of $Ni_{12}P_5$.

**Long-term stability evaluations**. To highlight the excellent stability of $Ni_{12}P_5$, a 100-h long-term test was conducted in a flow reactor system in which the $H_2$ and $CO_2$ gases were continuously flowed through the catalyst and an external heat input was adopted. As shown in Fig. 3c, under a light irradiation with intensity of 0.8 W cm$^{-2}$ and a controlled apparent temperature of 290 °C, the 10.4 wt% $Ni_{12}P_5/SiO_2$ sample afforded an initial CO production rate and selectivity of 13.5 mmol $g_{cat}^{-1}$ h$^{-1}$ and 99.9% during the first 10 h, respectively. respectively, and only a very small decrease to 13.3 mmol $g_{cat}^{-1}$ h$^{-1}$ and 99.8% was observed after 100-h of continuous testing, thereby demonstrating its excellent long-term stability. In comparison, a control $Ni/SiO_2$ sample lost 76% of its initial CO production rate after 100 h under similar conditions and declined to a final CO selectivity of about 85% (Supplementary Fig. 10).

In addition, in the same flow reactor tests were also performed at different temperatures with and without light illumination. As shown in Supplementary Fig. 11, under thermal activation (dark conditions), the 10.4 wt% $Ni_{12}P_5/SiO_2$ sample can afford a CO production rate of 8.03 mmol $g_{cat}^{-1}$ h$^{-1}$ at 320 °C. Furthermore, the CO production rate under light conditions (i.e., both thermal and photo activation) at the same apparent temperature was 25.6 mmol $g_{cat}^{-1}$ h$^{-1}$, corresponding to a 2.2-fold enhancement. This

enhancement was more significant at lower apparent temperature, which again indicated a solar advantage of the photothermal effect of the $Ni_{12}P_5$ catalyst.

The TEM images of the spent photocatalysts offered further evidence of the excellent anti-sintering ability of $Ni_{12}P_5/SiO_2$, which was conducive to its remarkable stability. As shown in Supplementary Fig. 12, the low-magnification image and particle size distribution of the $Ni_{12}P_5/SiO_2$ photocatalyst indicated that the overall mean particle size increased only slightly from about 13 nm to about 14 nm after 100 h of photocatalytic reaction. According to the corresponding HRTEM image, a high degree of crystallinity was well retained, implying the robust stability of the $Ni_{12}P_5$ catalyst, as well as its RWGS activity. In comparison, highly dispersed Ni metal nanoparticles of about 3.3 nm in diameter were found to have strikingly grown during long-term testing, which likely accounts for its considerable performance decay (Supplementary Fig. 13). The serious sintering of metallic Ni catalysts may be related to Mond process chemistry[25], in which the Ni first reacted with the product CO to form the $Ni(CO)_4$, and at higher temperature the $Ni(CO)_4$ decomposes back to larger Ni particles. The formation of $Ni(CO)_4$ was notably significantly suppressed on nickel phosphide[26,27], thus reducing the possibility of Mond process and consequent sintering.

**In situ DRIFTS studies**. In situ diffuse reflectance infrared Fourier transform spectroscopy (DRIFTS) measurements were conducted to investigate the reaction mechanism occurring at the $Ni_{12}P_5$ surface. DRIFTS spectra were acquired at 300 °C for 1 h in $CO_2/H_2$ flow after equilibrium, as presented in Fig. 4a. All DRIFT plots exhibited a pronounced peak centered at 2082 cm$^{-1}$ and a weak, broad peak centered at 1920 cm$^{-1}$, which could be assigned to linearly bonded $Ni^0$-CO and bridge-bonded $Ni^0$-CO, respectively[28,29]. The positions of the aforementioned two peaks are the same, regardless of the loading and size of the $Ni_{12}P_5$ particles, however, with a noticeable broadening and pronounced asymmetry of FTIR peak as the loading amount decreased, based on the FWHM analysis (Supplementary Table 4). This indicates that there is a range of surface sites for linearly bonded carbonyl species on the $Ni_{12}P_5$ particles, likely caused by the two distinct types of Ni site in $Ni_{12}P_5$ and their arrangements in different exposed facet[30,31]. Nevertheless, the intensity of the former peak overwhelms that of the latter, indicating that most carbonyl molecules preferred bonding individually to separated Ni atoms

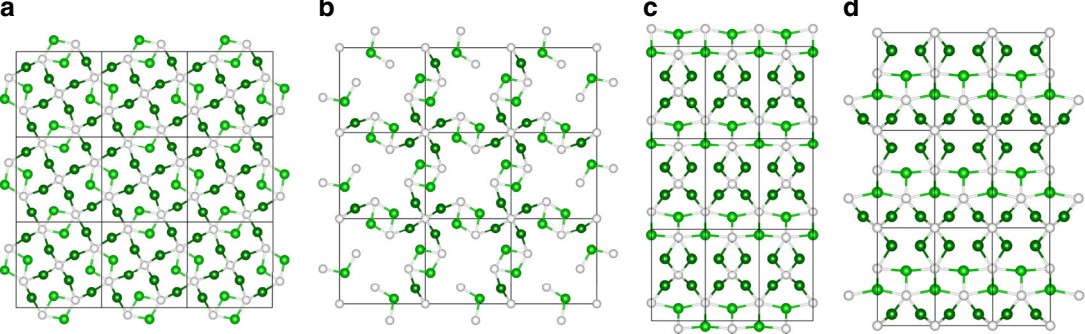

**Fig. 5 Surface crystal structure perspective of Ni$_{12}$P$_5$. a, b** In the (001) orientation, **c, d** in (010) orientation. Note the white spheres represent the P atoms, and Ni atom with two different coordination environments are depicted as dark green and light green spheres, respectively.

over engaging in multi-coordinate bonding. Regarding the metallic Ni on which the carbonyl molecule would be bonded in both linearly and bridging modes, observations based on the DRIFTS spectra of Ni$_{12}$P$_5$ implied the occurrence of unique, linearly bonded nickel-carbonyl-dominated reaction pathway. This can be explained by the ensemble effect[32–34], as in this case the P atoms in Ni$_{12}$P$_5$ effectively separate Ni atoms into highly dispersed nanoclusters (as depicted in the crystal structure of the Ni$_{12}$P$_5$ in Fig. 5). As aforementioned, EXAFS results concluded that bonding between Ni and P resulted in an increased Ni–Ni bond length, which could geometrically hamper carbonyl bridge-bonding[35,36]. In addition, the absence of the fingerprint modes of formate species (as indicated in Supplementary Fig. 14) implied that the reverse water gas shift reaction over Ni$_{12}$P$_5$ likely occurred through the direct CO$_2$ dissociation route[10], in accordance with reported DFT predictions[37].

The merit of linearly bonded carbonyl on Ni sites lies in the relative weaker binding energy when compared to those of bridge bonded species, and would facilitate the desorption of produced CO. In addition, the decreased coordination number of metal sites (i.e., Ni in Ni$_{12}$P$_5$) was deemed to interfere with the carbonyl dissociation and improved CO desorption[38]. Thereby, the consecutive reaction of CO with H$_2$ for methane production would be hindered on the Ni$_{12}$P$_5$ surface, leading to a significantly near 100% selectivity to the RWGS reaction.

## Discussion
By virtue of the well-isolated Ni nanoclusters, the linearly bonded nickel-carbonyl-dominated reaction pathway along with high RWGS reaction rate and selectivity over Ni$_{12}$P$_5$ were well preserved when changing the reaction conditions. As shown in Fig. 4b and Fig. 4c, altering the initial CO$_2$:H$_2$ ratio (between 5:1 and 1:5) or reaction temperature (between 160 °C and 320 °C) had little influence on the selectivity. As a comparison, many studies have demonstrated that the reaction conditions, such as initial CO$_2$/H$_2$ ratio, CO$_2$ conversion, temperature, catalyst particle size could significantly influence the CO$_2$ hydrogenation selectivity over nickel related catalysts. For example, it has been reported that a higher proportion of H$_2$ in the feed usually favors a high selectivity towards CH$_4$ production[37]. Moreover, increasing the conversion of CO$_2$ or raising the reaction temperature would also improve the selectivity to CH$_4$ on a nickel catalyst[39,40]. The size of the nickel catalyst also play an important role in the reaction selectivity, where a smaller particle size would favor the formation of CO rather than CH$_4$[10,40]. The size effect was more pronounced in the case of a Ni single atom catalyst, on which the CH$_4$ production was almost eliminated[41].

In stark contrast, the observations on Ni$_{12}$P$_5$ catalyst indicate that it can direct surface reaction pathways into the RWGS for

CO production, and the near 100% CO selectivity for the Ni$_{12}$P$_5$ catalyst was virtually independent of overall particle size, gas composition and reaction temperature, which is a distinctive feature among nickel-containing catalyst. This unusual property thus enables Ni$_{12}$P$_5$ to function as a distinctive class of RWGS catalyst and can be considered to extend Ni catalyst science and technology towards broader applicability, such as in tandem with other catalysts known for their capacity for CO conversion reactions[42].

Besides Ni$_{12}$P$_5$, the metal phosphide design concept was also demonstrated in the case of a cobalt phosphide analog. As shown in Supplementary Fig. 15, the as-prepared material can be well indexed to orthorhombic Co$_2$P. Like Ni$_{12}$P$_5$, it is also able to absorb a wide range of the solar spectrum. In order to increase its specific area, the SiO$_2$ supported Co$_2$P was also synthesized, successfully shrinking the particle size to around 14 nm (Supplementary Fig. 16). Photocatalytic reactions over Co$_2$P showcased a CO production rate of 15.7 mmol g$_{cat}^{-1}$ h$^{-1}$ that was maintained for at least six usage cycles, and which could be further boosted to 227.7 mmol g$_{cat}^{-1}$ h$^{-1}$ after loading onto SiO$_2$ (Supplementary Fig. 17). It is worth noting that the selectivity of Co$_2$P toward CO was still quite high (over 99%), and exhibited the same independence of overall particle size and CO$_2$/H$_2$ reactant ratio effects. Similar to Ni$_{12}$P$_5$, the crystalline structure of Co$_2$P also displayed cobalt nanoclusters that were well-separated by P atoms (Supplementary Fig. 15d). Thus, the results presented herein over the Co$_2$P photocatalyst have served to further validate the metal phosphide design philosophy.

In summary, the photothermal catalytic capability of Ni$_{12}$P$_5$ has been unlocked to drive a highly efficient reverse water gas shift reaction using simulated solar light, with a peak CO production rate of 960 ± 12 mmol g$_{cat}^{-1}$ h$^{-1}$ and selectivity consistently greater than 99.5%. The key to this performance lies in the ensemble effect of P, which results in the highly dispersed Ni nanoclusters in Ni$_{12}$P$_5$, and disfavored strong multi-coordinate bonding to CO. In addition, the excellent light-harvesting ability of Ni$_{12}$P$_5$ across the entire solar spectrum also contributed to its photothermal activity, thereby enabling a considerably elevated local temperature to drive the RWGS reaction. Importantly, the advantages of high catalytic activity and light capture are not exclusive to Ni$_{12}$P$_5$, being shared by other transition metal phosphides such as Co$_2$P. Therefore, as a class of high-performance, stable and cost-effective materials, the transition metal phosphides offer interesting opportunities for the development of photothermal CO$_2$ conversion technologies.

## Methods
**Synthesis of Ni$_{12}$P$_5$ and SiO$_2$-supported Ni$_{12}$P$_5$.** A temperature programmed reduction (TPR) method was adopted to prepare the Ni$_{12}$P$_5$ nanoparticles. In detail, nickel nitrate and ammonium phosphate, (NH$_4$)$_2$HPO$_4$, with molar ratio of

12:5 were first dissolved in DI water to form the precursor solution, and then dried at 120 °C, followed by annealing at 550 °C for 6 h to form nickel phosphate oxide. This precursor solid was then reduced under flowing 10 % $H_2$/Ar at 600 °C for 6 h (with ramp of 10 °C min$^{-1}$ from 25 °C to 300 °C and then 2 °C min$^{-1}$ to 600 °C). The series of $SiO_2$-supported $Ni_{12}P_5$ with various catalyst loadings was prepared via incipient wetness impregnation method. In a typical synthesize, the $SiO_2$ was first immersed into a certain volume of the aforementioned precursor solution, and then followed by the same dry and $H_2$-treatment process of the $Ni_{12}P_5$ synthesis. In addition, a Ni/$SiO_2$ reference was also prepared in a similar impregnation route except the ammonium phosphate was not introduced into the precursor solution, and the $H_2$-treament temperature was 350 °C.

**Material characterization.** Powder X-ray diffraction patterns were recorded on a Bruker D2 Phaser X-ray diffractometer. UV-visible-NIR diffuse reflectance spectra were measured on Lambda 1050 UV/Vis/NIR spectrometer from Perkin Elmer, equipped with an integrating sphere. Low-resolution TEM images were taken on a FEI T12 G2 and high-resolution TEM images were taken on a Hitachi H-7650 HR-TEM. X-ray photoelectron spectroscopy (XPS) was performed on a Perkin Elmer Phi 5500 ESCA spectrometer in an ultra-high vacuum chamber, with all results being calibrated to the C 1s peaks at 284.5 eV. An Optima 7300DV ICP-OES apparatus (Perkin Elmer, USA) was used to measure the weight percent the $Ni_{12}P_5$ content in $Ni_{12}P_5$/$SiO_2$ samples. Pulse $H_2$ chemisorption was performed on AutoChem II 2920 to detect the metal dispersion on each sample, assuming atomic hydrogen only binds to surface nickel atoms with a Ni/H = 1.

X-ray absorption spectroscopy measurements were performed at Sector 9-BM of the Advanced Photon Source at Argonne National Laboratory (Lemont, IL) using a Quick-EXAFS monochromator and gas-ionization chamber detectors. A specially designed sample holder/chamber was used to provide simultaneous control of atmospheric composition and catalyst temperature during in situ measurements[43]. The $Ni_{12}P_5$ material was sealed in the sample chamber, purged with He and heated to 300 °C before flowing a 5:1 ratio of $CO_2$/$H_2$ and acquiring EXAFS spectra at 30 s intervals. XANES data processing was performed using Athena, part of the Demeter software package[44]. Fitting of EXAFS spectra was performed using WinXAS software[45], in conjunction with scattering paths generated using FEFF8[46] and based on crystal structures obtained from the Crystallographic Open Database[47]. The loading amount were calculated using ICP-AES data acquired on a Thermo iCAP 6300 spectrometer system.

**Photocatalytic gas-phase $CO_2$ reduction tests.** A batch reactor system was adopted to test the photocatalytic performance of the materials (detailed setup is shown in Supplementary Fig. 18). Before photocatalytic testing, 0.5 mg of the sample was well-dispersed in DI water via sonication and then drop-cast onto a binder-free borosilicate glass microfiber filter with an area of 1 cm$^2$ before being dried under vacuum at 80 °C. The filter-supported sample was then fixed into a custom-fabricated 11.8 mL stainless steel batch reactor with a fused silica window and finally sealed with a Viton O-ring. The reactor was then evacuated before being sequentially filled with $H_2$ and $CO_2$ gas. The total pressure of the reactants was 18 psi (about 1.2 bar), with $P(H_2) = 3$ psi and $P(CO_2) = 15$ psi, as monitored by an Omega PX309 pressure transducer. An unfiltered 300 W Xe lamp was employed as the light source, and the light intensity was calibrated to 2.3 W with a light meter and a mask with an area of 1 cm$^2$. No external heating was adopted in batch reactor. Photocatalytic reactions were conducted with a duration of 0.5 h for each run, after which the product gases were separated on an SRI-8610 gas chromatograph equipped with 30 Mole Sieve 13a and 60 Haysep D columns, and then analyzed using a flame ionization detector (FID) and thermal conductivity detector (TCD). For recyclable tests, the reactor was evacuated and refilled with the reactant gases after each run, without any treatments or air (oxygen) exposure. For tests under different light intensity and wavelength range, a series neutral-density (ND) filters and high pass cut-off filters were adopted, respectively. The $^{13}C$ isotopic labeling experiments were also conducted in the same reactor, with the $^{12}CO_2$ being replaced by $^{13}CO_2$ and the products being measured on an Agilent 7890 A gas chromatograph/mass spectrometer equipped with a 60 m GS-CarbonPLOT column.

**Long-term photocatalytic gas-phase $CO_2$ reduction tests.** A flow reactor system was used for the long-term stability evaluation on the photocatalysts (detailed setup is shown in Supplementary Fig. 19). Firstly, the sample was packed into a tubular quartz capillary reactor with an inner diameter of 2 mm and an outer diameter of 3 mm. A gas flow of $CO_2$ (2.5 sccm) and $H_2$ (0.5 sccm) was then introduced into the reactor using Alicat Scientific digital flow controllers, and the temperature was regulated by an OMEGA CN616A temperature controller. The gas products were analyzed automatically on an SRI-8610, as described previously for photocatalytic activity measurements.

**In situ DRIFTS measurements.** In situ DRIFTS experiments were conducted on a Thermo Scientific™ Nicolet™ iS50 FT-IR Spectrometer with a HgCdTe detector. A Harrick Praying Mantis™ diffuse reflection accessory and a Harrick high-temperature reaction chamber (HTC) with ZnSe windows were employed, with an additional Harrick ATC-024-3 controller used to provide temperature control. The

samples were sealed into the sample cup and then heated to 350 °C under He flow (20 sccm) for 2 h. The samples were then cooled down to room temperature, during which time a series of background spectra were collected at different temperatures. Next, a mixture of $CO_2$, $H_2$ and He gases was purged into the chamber using flow rates of 5, 1 and 14 sccm, respectively. Finally, spectra were collected by taking 128 scans at each different temperature stage with a resolution of 4 cm$^{-1}$.

**Synthesis of $Co_2P$ and $SiO_2$-supported $Co_2P$.** The synthetic route used to produce $Co_2P$ is similar to that used for $Ni_{12}P_5$. Cobalt nitrate and $(NH_4)_2HPO_4$ were first dissolved in DI water at a molar ratio of 2:1, then dried and calcined at 120 °C and 550 °C, respectively. These precursor solids were finally reduced in 10% $H_2$/Ar flow at 720 °C for 6 h (with a ramp rate of 2 °C min$^{-1}$ from 25 °C to 720 °C). Again, an incipient wetness impregnation route was adopted to load $Co_2P$ onto the $SiO_2$ support material.

## Data availability
The data that support the plots within this paper and other findings of this study are available from the corresponding author upon reasonable request.

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

## Acknowledgements

G.A.O. acknowledges the financial support of the following agencies: Ontario Ministry of Research and Innovation; Ministry of Economic Development, Employment and Infrastructure; Ministry of the Environment and Climate Change; Best in Science; Ministry of Research Innovation and Science Low Carbon Innovation Fund; Ontario Centre of Excellence Solutions 2030 Challenge Fund; Alexander von Humboldt Foundation; Imperial Oil; University of Toronto Connaught Innovation Fund; Connaught Global Challenge Fund and the Natural Sciences and Engineering Research Council of Canada. D.K. acknowledges the financial support from the National Natural Science Foundation of China (21875288), the GDUPS (2016). This research used resources of the Advanced Photon Source, a U.S. Department of Energy (DOE) Office of Science User Facility operated for the DOE Office of Science by Argonne National Laboratory under Contract No. DE-AC02-06CH11357. Sector 9-BM beamline scientists Tianpin Wu and George Sterbinski of the APS Spectroscopy group are thanked for their expert assistance in beamline optimization and operation. P.N.D. acknowledges financial support from the NSERC Postdoctoral Fellowships program. A. Tountas' help on ASPEN estimation is acknowledged by all authors. This paper is dedicated to Professor Geoffrey Ozin on the occasion of his 77th birthday.

## Author contributions

Y.X. and G.A.O. conceived and designed the experiments. Y.X. prepared the materials and performed the SEM, XRD, UV-vis-NIR, in situ DRIFTS and photocatalytic characterizations. L.W. performed XPS characterizations. F.M.A. and J.L. performed the TEM characterizations. M.X. performed the ICP-OES detection. A.A.T. conducted the ASPEN simulation. P.N.D. performed and analyzed the XAS experiments. J.L. and D.K. performed the pulse $H_2$ chemisorption tests. Y.X., P.N.D. and G.A.O. co-wrote the manuscript. All authors discussed the results and commented on the manuscript.

## Competing interests

The authors declare no competing interests.
