## [Peer Review File · Nature Communications]

Reviewers' comments:

Reviewer #1 (Remarks to the Author):

The authors present evidence of high RWGS activity and selectivity for a nickel phosphide catalyst (Ni₁₂P₅/SiO₂) used in the photothermal CO₂ hydrogenation reaction – a very hot research topic currently. It is argued that Ni₁₂P₅ nanoparticles in the catalysts provide atom-precise, structurally uniform Ni sites for the reaction that enable high activity and selectivity that is not achieved for a Ni metal catalyst (Ni/SiO₂). Promising RWGS properties for a cobalt phosphide catalyst (Co₂P/SiO₂) are also reported and the authors suggest that the results provide additional evidence for atom-precise catalysis by metal phosphides.

The strength of the manuscript lies in the observation of high activity and nearly 100% selectivity towards CO for the CO₂ hydrogenation reaction over Ni₁₂P₅/SiO₂ catalysts exposed to a wide-band light source, and the evidence presented to support a hypothesis that photothermal catalysis is operative under the reaction conditions used. To the reviewer the result presented are provocative but not yet convincing. If the results hold up, the findings suggest that a relatively easily prepared, earth abundant metal phosphide dispersed as large-ish nanoparticles on silica is a highly promising catalyst for photothermal conversion of CO₂ to CO. However, the reviewer notes a number of weaknesses with the manuscript, and these would need to be addressed to raise needed confidence in the results, their analysis, and presentation. Weaknesses in the manuscript identified by the reviewer are listed below:

1. Title: The title of the manuscript is misleading for two reasons – nickel phosphide is not mentioned, and the idea that Ni₁₂P₅ nanoparticles provide a nanocluster superlattice is not well supported.
2. Page 1 and Figure 1: As with the title, this section about nanoclusters in heterogeneous catalysis seems disconnected from the results of the study. As discussed below, there is no strong evidence that the Ni₁₂P₅ nanoparticles (in bulk or silica supported form) provide “atomic and crystalline perfection” as catalytic sites.
3. Page 3, paragraphs 1 and 3: Were atomic compositions (i.e. Ni_xP_y) determined by ICP-OES and XPS? It is critical that this information be provided in the manuscript. TPR-prepared metal phosphides, including nickel phosphides, typically have bulk and surface compositions that differ from the expected stoichiometric compositions (e.g., see: Journal of Catalysis 231 (2005) 300–313). This is due to the loss of some P (as PH₃) during synthesis and can lead to P-poor and P-rich materials (and surfaces) despite high crystallinity identified by XRD and TEM. This is important with regard to thinking of nickel phosphides as atomically-precise crystalline materials for catalysis.
4. TEM results: The average particle diameters for the bulk and supported Ni₁₂P₅ particles are large, as are the standard deviations. The wide particle size distributions indicate that a range of surface facets are exposed on the Ni₁₂P₅ particles for all loadings, resulting in a broad variety of surface geometries and sites available to adsorbed species.
5. Further to Point #3, a range of characterization methods, including ³¹P NMR and Mossbauer spectroscopy, indicate a range of metal and P environments in nanoscale metal phosphide particles (e.g., see: J. Catal. 2010, 276, 249-258; J. Catal. 2010, 272, 18-27; Catal. Lett. 2012, 142, 1413-1436). The authors need to consider these and similar published results (along with comments #3 & 4), if they are to build a case that the atom-precise structure of Ni₁₂P₅ in their catalysts is responsible for the high photothermal catalytic activity observed.
6. Page 4, top (EXAFS): The authors should consider the detailed EXAFS results and analyses of the research groups of Oyama and Prins in their analysis of the EXAFS data presented in the manuscript.
7. Photocatalytic Performance section and relevant Methods section: It is unclear to the reviewer if,

when, and how much external heating of the catalyst was applied during the photocatalytic activity measurements.

8. Page 5, top (...the turnover frequency of Ni₁₂P₅...) The suggestion that the decrease in TOF with decreasing Ni₁₂P₅ loading is due to "poorer light harvesting ability and associated photothermal effect" is not supported by results, and the reviewer is quite skeptical of this conclusion.

9. Local temperature estimation: More information about how the local temperatures were estimated using ASPEN software would be helpful – this could be included in the supporting information. The authors should add error bars to the estimated local temperatures.

10. "Unprecedented": This word is used in three places to describe the high RWGS activity measured for the Ni₁₂P₅ catalysts. The basis for the "unprecedented" description is not explained, and it isn't clear how such a conclusion can be made without comparing the currently reported RWGS activities to all others reported in the literature.

11. Page 7, top (flow reactor measurements): The flow reactor measurements were conducted over the high-activity Ni₁₂P₅/SiO₂ catalyst at 290 C, yet much lower activities were measured than in the batch reactor measurements. If the reviewer is interpreting correctly, the activities measured in the batch reactor at 290 C under light illumination are nearly two orders of magnitude lower than in the batch reactor study. Why is this, and how do the flow reactor activities compare with relevant data in the literature. These results highlight how photocatalytic activities reported in the literature tend to be very dependent on the reaction conditions used, making comparisons difficult. The authors should comment on this.

12. Page 7, sintering of Ni/SiO₂: The dramatic sintering of the Ni/SiO₂ catalyst seems surprising under the reaction conditions used. The authors should report the temperature of the catalyst during the measurements. The sintering might be due to the formation of Ni(CO)₄ under reaction conditions, thus facilitating transport of Ni for particle growth. Transmission IR spectra of CO on Ni/SiO₂, Ni₁₂P₅/SiO₂ and Ni₂P/SiO₂ catalysts have been reported previously (e.g., *Journal of Catalysis* 231 (2005) 300–313) and it was shown that Ni(CO)₄ formed on Ni/SiO₂, but the formation of this species was suppressed on the Ni phosphide catalysts. Another relevant IR study is the following: *J. Phys. Chem. B* 2004, 108, 10930-10941.

13. Figure 5a: The IR peak for terminally-bonded CO noticeably broadens as the Ni₁₂P₅ loading decreases. Not surprisingly, this suggests that there are a range of CO bonding environments on the Ni₁₂P₅ particles and that this becomes more pronounced for smaller particles. The authors should discuss the relevancy of this to the proposal of Ni₁₂P₅ providing "atom-precise" catalytic sites.

14. Page 8, top (IR spectral interpretation): The authors should cite relevant literature that have assigned and interpreted the IR spectra of adsorbed CO on Ni phosphide catalysts.

15. Page 8, middle (Compared to metallic Ni,...): It is suggested that the selectivity of metallic Ni or single-atom Ni catalysts "would change a lot" under the reaction conditions probed – the authors should be specific in making this statement.

16. Page 8, bottom: For the Ni₁₂P₅ particle range studied (7.8-13.3 nm), the distribution of exposed facets doesn't change much with particle size as the biggest changes happen for particle diameters below ~5 nm.

17. Page 9, top: The sentence ending with "such as employing it in tandem with other catalysts known for their capacity for CO conversion reactions" is unclear to the reviewer. The authors should explain this more fully.

18. Page 9, top: The sentence "This atom-perfect design philosophy is expected to be applicable to other compounds..." is quite strong and needs to be considered in light of the comments above, and more evidence for atom-perfect design provided if it is to be included in the manuscript.

Reviewer #2 (Remarks to the Author):

This work describes the use of Ni₁₂P₅ and its supported form as an active photothermal catalyst for RWGS. The manuscript is clearly written and their good activity and high stability have been highlighted. The materials are also well characterized. After going through the details; however, I evaluate that the novelties and impacts of the contents are not sufficiently high for the journal under evaluation. Therefore, I do not recommend this manuscript for publication in Nature Communications. Some detailed comments are listed below.

- The term "photocatalysis" in the title is somewhat misleading. Probably much more precise to call it "photothermal catalysis".
- What is the reason for the decreased absorbance of Ni₁₂P₅/SiO₂ at longer wavelength? Can the UV-Vis spectrum of SiO₂ included in the SI?
- Regarding the TEM study, is it reasonable and justified to report so accurate numbers with the significant figures (e.g. 86.1±30.2 nm and 0.604 nm and 0.401 nm)? The same holds for other values (e.g. EXAFS: 0.163 nm and 0.264 nm).
- Similarly, selectivity is shown up to 1 decimal point. Can it be so accurate? Standard deviation?
- What are other products observed? Methane?
- Fig. 1 is beautiful, but I do not see much the relevance in this work.
- Why were 3 psi H₂ and 15 psi CO₂ and the specific ratio chosen? Pressure unit should be changed.
- Conversion of gases and equilibrium conversions should be explicitly mentioned.
- Regarding the estimation of the catalyst temperature, the equilibrium conversion at different temperatures was used. This is only possible when the conversion is very high and at the equilibrium (according to the text, it is the case). It would be important to show the temporal evolution of the reaction performance for one or a few specific case(s) so that one can indeed reach the equilibrium conversion.
- Catalytic performance under thermal activation should be reported.
- Regarding the catalytic tests performed under continuous flow (Figs. 4c and 5d), it says "controlled apparent temperature of 290 °C" for the former. This means that the sample heated? Similar to above, what is the catalytic activity without illumination?
- The details of the setup (light source, focus, etc.) should be described ideally with photograph of the setup.

Reviewer #3 (Remarks to the Author):

The authors presented a novel application of metal phosphide based catalyst for photocatalytic RWGS applications. This is an important piece of work which would help to boost the study of transitional metal phosphide in critical industrial processes. However, several issues need to be rectified before the manuscript is ready for publications.

1. It is unclear from the work if the RWGS by Ni₁₂P₅ is photocatalytic or simply a photothermal effect.
2. Is this a promising approach to achieve sustainable RWGS reaction? What is the performance of the supported Ni₁₂P₅ catalyst compared to other reported RWGS catalysts?
3. I find it hard to discern from Figure 3c if there is any change to the presented in-situ XANES

spectra.

4. Why was a lower light intensity used for the long term stability testing in a flow reactor? From the results presented, the performance of the catalyst was reduced by over 100 times. Is the stability of the catalyst reproducible under the same illumination intensity as the batch reaction?

5. The authors state that the observed performance can be attributed in the periodic array of uniform and atom-precise Ni nanoclusters in Ni₁₂P₅, a nanocluster superlattice catalyst, which disfavors strong multi-coordinate bonding to CO. However, the nature of the Ni₁₂P₅ active site(s) during the catalytic reaction is not clear. Is TEM alone sufficient to support the retainment of the Ni₁₂P₅ structure after the catalytic reaction?

6. Corresponding UV-Vis-NIR spectra would be helpful to support the authors' attribution of the decreasing TOF with lower Ni₁₂P₅ loading to poorer light harvesting ability. Nonetheless, a performance test at the same temperature in the absence of light would be a better experiment to prove the authors' idea.

7. Additional comments:

a. The following sentence is too convoluted:

Combined with the aforementioned photocatalytic tests performed on Ni₁₂P₅ with varying particle diameters, these observations indicate that the nearly 100% CO selectivity of the Ni₁₂P₅ catalyst was virtually independent of overall particle size, the gas composition and the reaction temperature, in other words, beyond the prevailing view of the size effect.

b. The discussion section only consists of a summary of the work.

c. Poor selection of significant figures, for example: 960.3±11.7 mmol gcat-1 h-1

d. Missing punctuation:

The positions of the aforementioned two peaks are the same, regardless of the loading and size of the Ni₁₂P₅ particles. The intensity of the former peak overwhelms that of the latter, indicating that most CO molecules preferred bonding individually to separated Ni atoms over engaging in multi-coordinate bonding.

Response to the Reviewers' Comments:

Reviewer #1 (Remarks to the Author):

The authors present evidence of high RWGS activity and selectivity for a nickel phosphide catalyst ($\text{Ni}_{12}\text{P}_5/\text{SiO}_2$) used in the photothermal CO_2 hydrogenation reaction – a very hot research topic currently. It is argued that Ni_{12}P_5 nanoparticles in the catalysts provide atom-precise, structurally uniform Ni sites for the reaction that enable high activity and selectivity that is not achieved for a Ni metal catalyst (Ni/SiO_2). Promising RWGS properties for a cobalt phosphide catalyst ($\text{Co}_2\text{P}/\text{SiO}_2$) are also reported and the authors suggest that the results provide additional evidence for atom-precise catalysis by metal phosphides.

The strength of the manuscript lies in the observation of high activity and nearly 100% selectivity towards CO for the CO_2 hydrogenation reaction over $\text{Ni}_{12}\text{P}_5/\text{SiO}_2$ catalysts exposed to a wide-band light source, and the evidence presented to support a hypothesis that photothermal catalysis is operative under the reaction conditions used. To the reviewer the result presented are provocative but not yet convincing. If the results hold up, the findings suggest that a relatively easily prepared, earth abundant metal phosphide dispersed as large-ish nanoparticles on silica is a highly promising catalyst for photothermal conversion of CO_2 to CO. However, the reviewer notes a number of weaknesses with the manuscript, and these would need to be addressed to raise needed confidence in the results, their analysis, and presentation. Weaknesses in the manuscript identified by the reviewer are listed below:

Re: Greatly appreciate all of the reviewer's constructive and positive comments.

1. Title: The title of the manuscript is misleading for two reasons – nickel phosphide is not mentioned, and the idea that Ni_{12}P_5 nanoparticles provide a nanocluster superlattice is not well supported.

Re: Appreciate the reviewer's comments, the title has been changed to "High-Performance Light-Driven Heterogeneous CO_2 Catalysis with Near-Unity Selectivity on Metal Phosphides"

2. Page 1 and Figure 1: As with the title, this section about nanoclusters in heterogeneous catalysis seems disconnected from the results of the study. As discussed below, there is no strong evidence that the Ni_{12}P_5 nanoparticles (in bulk or silica supported form) provide "atomic and crystalline perfection" as catalytic sites.

Re: Appreciate the reviewer's comments. We understand the reviewer's concern. The crystal structure of Ni_{12}P_5 defines nickel clusters composed of few Ni atoms integrated into a P lattice. These nanoclusters have translational periodicity in the crystal and can be described as a nanocluster superlattice, namely a lattice based upon a periodic array of nickel nanoclusters. No matter which facet is exposed in Ni_{12}P_5 nanocrystals a periodic array of these nickel nanoclusters exists on the surface of the nanocrystals and act as catalytic sites, no matter what the state of their dispersity (as

depicted in Figure R1). The following experimental findings support this point of view:

- a) **The near-unity selectivity towards CO production.** With regard to literature reports on conventional Ni catalysts, high CO selectivity was only observed on ultrasmall Ni clusters or single atom catalysts, otherwise CO₂ methanation would dominate.^{1,2}
- b) **Linear bonded of CO on the surface during catalysis as revealed by in-situ DRIFT spectra.** This observation indicates a single-site reaction mechanism.
- c) **The aforementioned proposal of an atomically precise nickel cluster superlattice in Ni₁₂P₅ is deemed responsible for the near-unity selectivity to CO production.** We agree surface defects could be responsible for the tiny amount of CH₄ by-product during the catalysis.

We have revised the paper accordingly

- a) The introductory statement related to “atomic perfection” has been modified
- b) Fig. 1 has been greatly improved to support the thesis
- c) In the discussion, we clarify the origin of the near-unity selectivity of CO production during photocatalysis where a tiny amount of CH₄ by-product originates from surface imperfections.
- d) The surface crystal structure illustration (Figure R1) was added in the revised manuscript as shown in Fig. 5d-g

Figure R1. Surface crystal structure perspective of Ni₁₂P₅ in the (001) orientation (a-b) and (010) orientation (c-d). Note the white spheres represent the P atoms, and Ni atom with two different coordination environments are depicted as dark green and light green spheres, respectively.

References:

- [1] Winter, L. R.; Gomez, E.; Yan, B. H.; Yao, S. Y.; Chen, J. G. G. *Appl. Catal. B* **2018**, 224, 442-450.
- [2] Millet, M. M.; Algara-Siller, G.; Wrabetz, S.; Mazheika, A.; Girgsdies, F.; Teschner, D.; Seitz, F.; Tarasov, A.; Leychenko, S. V.; Schlögl, R.; Frei, E. *J. Am. Chem. Soc.* **2019**, 141, 2451-2461.

3. Page 3, paragraphs 1 and 3: Were atomic compositions (i.e. Ni_xP_y) determined by ICP-OES and XPS? It is critical that this information be provided in the manuscript. TPR-prepared metal phosphides, including nickel phosphides, typically have bulk and surface compositions that differ

from the expected stoichiometric compositions (e.g., see: Journal of Catalysis 231 (2005) 300–313). This is due to the loss of some P (as PH₃) during synthesis and can lead to P-poor and P-rich materials (and surfaces) despite high crystallinity identified by XRD and TEM. This is important with regard to thinking of nickel phosphides as atomically-precise crystalline materials for catalysis.

Re: Appreciate the reviewer's query. The atomic compositions presented in this work were checked by ICP-OES and show the Ni/P ratio in the as-prepared Ni₁₂P₅ and Ni₁₂P₅/SiO₂ samples were 2.26-2.37, quite similar to the stoichiometric ratio of 2.4.

The reviewer is correct that the element composition, especially at the nanocrystal surface could deviate from stoichiometric. Surface imperfections and dispersions always exist in real nanomaterials. Nevertheless, as mentioned above, the near unity CO selectivity can be explained in terms of the atomic perfection of the nickel cluster superlattice while surface imperfections yield the CH₄ byproduct. This point has been clarified throughout the revised version of the paper.

4. TEM results: The average particle diameters for the bulk and supported Ni₁₂P₅ particles are large, as are the standard deviations. The wide particle size distributions indicate that a range of surface facets are exposed on the Ni₁₂P₅ particles for all loadings, resulting in a broad variety of surface geometries and sites available to adsorbed species.

Re: Appreciate the reviewer's careful reading. In this work, the Ni₁₂P₅ particle size was controlled by the loading, where a lower loading would give rise to a smaller particle size. We concur there exists a particle size distribution but as argued above the same atom-precise nickel nanoclusters are present in every facet for every size and shape nanoparticle. The in-situ DRIFT results point to single-site reaction behavior for all Ni₁₂P₅/SiO₂ samples no matter the state of dispersion. In addition, in contrast to well-studied literature examples of metallic nickel, the CO₂ hydrogenation reaction selectivity over Ni₁₂P₅ catalyst did not show a dependence on the particle size, which is always near 100% selectivity towards the RWGS reaction. Support for the above findings emerged from X-ray absorption spectroscopy, XAS, measurements. The XAS results indicate that the increased Ni-Ni bond length and 'Ni cluster P ensemble effect' are likely responsible for the unique Ni₁₂P₅ hydrogenation behavior towards CO₂. As mentioned, these effects would be invariant to the nanocrystal dispersion, hence we consider dispersion differences would not much influence surface adsorbed species and reaction pathways.

5. Further to Point #3, a range of characterization methods, including ³¹P NMR and Mossbauer spectroscopy, indicate a range of metal and P environments in nanoscale metal phosphide particles (e.g., see: J. Catal. 2010, 276, 249-258; J. Catal. 2010, 272, 18-27; Catal. Lett. 2012, 142, 1413-1436). The authors need to consider these and similar published results (along with comments #3 & 4), if they are to build a case that the atom-precise structure of Ni₁₂P₅ in their catalysts is responsible for the high photothermal catalytic activity observed.

Re: Appreciate the reviewer's reading list of relevant papers and based on the aforementioned proposals we have clarified this point throughout the revised version of the paper.

6. Page 4, top (EXAFS): The authors should consider the detailed EXAFS results and analyses of the research groups of Oyama and Prins in their analysis of the EXAFS data presented in the manuscript.

Re: Appreciate the suggestion. We have carefully compared our results with those presented in Oyama's or Prins's papers. The EXAFS spectra and fitting results in our study are in accordance with theirs, a point clarified in the revised version of the paper in which the related references are cited.

7. Photocatalytic Performance section and relevant Methods section: It is unclear to the reviewer if, when, and how much external heating of the catalyst was applied during the photocatalytic activity measurements.

Re: The tests in our batch reactor were conducted without external heating, while for those in our flow reactor, the temperature was achieved with the assistance of light (photothermal effect) and external heating, and controlled with a thermocouple probe. This has been clarified in the methods and discussion sections of the revised manuscript.

8. Page 5, top (...the turnover frequency of Ni_{12}P_5 ...) The suggestion that the decrease in TOF with decreasing Ni_{12}P_5 loading is due to "poorer light harvesting ability and associated photothermal effect" is not supported by results, and the reviewer is quite skeptical of this conclusion.

Re: The lower light harvesting efficiency of supported Ni_{12}P_5 was demonstrated by measuring the UV-vis-NIR spectra of all the samples. As shown below, as the loading amount decreased, the A% decreased especially in the NIR region, indicating the lower light harvesting ability of the lower loading samples. The new UV-vis-NIR spectra have been added into the revised manuscript as Fig. 2b.

Figure R2. UV-vis-NIR spectra of Ni_{12}P_5 , SiO_2 and Ni_{12}P_5 - SiO_2 samples loaded onto a binder-free borosilicate glass microfiber filter (0.5 mg/cm^2).

9. Local temperature estimation: More information about how the local temperatures were estimated using ASPEN software would be helpful – this could be included in the supporting information. The authors should add error bars to the estimated local temperatures.

Re: The accuracy of the ASPEN simulation depends on 1) the validity of the ideal gas approximation 2) if water stays in the gas phase for the duration of the experiment. Typically, we assume that 1) & 2) are met. 1) is valid for high temperature (>100 °C) and low-pressure systems (<2 atm).

For the estimation we used the ASPEN NRTL property package (assumes ideal gas phase) and the Gibbs reactor block that assumes all reaction between components (H₂, CO₂, CO, H₂O). A sweep was performed in 50 °C increments and the local temperature was determined by matching our GC results with the temperature dependent ASPEN output.

These points have been clarified in the revised version, please see the caption of the Supplementary Table 3.

10. “Unprecedented”: This word is used in three places do describe the high RWGS activity measured for the Ni₁₂P₅ catalysts. The basis for the “unprecedented” description is not explained, and it isn’t clear how such a conclusion can be made without comparing the currently reported RWGS activities to all others reported in the literature.

Re: In response to the reviewer’s comment, we have summarized the photothermal CO₂ hydrogenation performances of some reported catalysts. As seen in Table R1, the Ni₁₂P₅ and Ni₁₂P₅/SiO₂ in this work display a superior CO₂ conversion rate compared to this library of metal oxides and noble metal catalysts. The selectivity to CO in the CO₂ hydrogenation reaction over Ni₁₂P₅ and Ni₁₂P₅/SiO₂ was near 100%, quite distinct to the other Ni containing catalysts in the Table, which show a greater tendency towards CH₄ formation. This Table has been added into the revised manuscript as Supplementary Table 2.

Table R1. Comparison of CO₂ conversion rates for some CO₂ hydrogenation catalysts under light irradiation without external heating.

Catalyst	Light source	Feed composition	CO ₂ conversion rate	Selectivity	
				CO	CH ₄
Cu ₂ O ¹	300 W Xe light (full spectrum, 40 suns)	CO ₂ /H ₂ =83/17	70.3 mmol g ⁻¹ h ⁻¹	~100	
In ₂ O _{3-x} ²	300 W Xe light (~20 suns)	CO ₂ /H ₂ =50/50	238.8 mmol g ⁻¹ h ⁻¹	~100	
Pt/NaTaO ₃ ³	300 W UV-enhanced Xe lamp	CO ₂ /H ₂ =50/50	140.5 μmol g ⁻¹ h ⁻¹	99	1
Pd@Nb ₂ O ₅ ⁴	300 W Xe lamp	CO ₂ /H ₂ =50/50	1.8 m mol g ⁻¹ h ⁻¹	100	
Cu/Pd/H _y WO _{3-x} ⁵	300 W Xe lamp (1 W cm ⁻²)	CO ₂ /H ₂ =50/50	40.8 μmol g ⁻¹ h ⁻¹	100	
Fe@C ⁶	300 W Xe lamp	CO ₂ /H ₂ =50/50	26.1 mmol g ⁻¹ h ⁻¹	100	

FeO–CeO ₂ ⁷	300 W Xe lamp (2.2 W cm ⁻²)	CO ₂ /H ₂ /Ar=15/60/25	20 mmol g ⁻¹ h ⁻¹	97- 99.9%	
SA Ni/Y ₂ O ₃ ⁸	ambient daytime sunlight (from 0.52 to 0.7 kW m ⁻²)	CO ₂ /H ₂ /N ₂ =2.5/10/87. 5	7.5 L m ⁻² h ⁻¹		100
Ni/SiO ₂ ·Al ₂ O ₃ ⁹	solar simulator	CO ₂ /H ₂ /N ₂ =15/70/15	14.4 mmol g ⁻¹ h ⁻¹	2.8	97.2
NiO ⁹	solar simulator	CO ₂ /H ₂ /N ₂ =15/70/15	13.3 mmol g ⁻¹ h ⁻¹		100
Ni ₁₂ P ₅ (this work)	300 W Xe lamp (2.3 W cm ⁻²)	CO ₂ /H ₂ =83/17	155.7 mmol g ⁻¹ h ⁻¹	99.5	0.5
Ni ₁₂ P ₅ /SiO ₂ (this work)	300 W Xe lamp (2.3 W cm ⁻²)	CO ₂ /H ₂ =83/17	960.3 mmol g ⁻¹ h ⁻¹	99.7	0.3
Co ₂ P (this work)	300 W Xe lamp (2.3 W cm ⁻²)	CO ₂ /H ₂ =83/17	15.7 mmol g ⁻¹ h ⁻¹	99.2	0.8
Co ₂ P/SiO ₂ (this work)	300 W Xe lamp (2.3 W cm ⁻²)	CO ₂ /H ₂ =83/17	227.7 mmol g ⁻¹ h ⁻¹	99.5	0.5

References:

- [1] Wan, L. L.; Zhou, Q. X.; Wang, X.; Wood, T. E.; Wang, L.; Duchesne, P. N.; Guo, J. L.; Yan, X. L.; Xia, M. K.; Lie, Y. F.; Jelle, A. A.; Ulmer, U.; Jia, J.; Li, T.; Sun, W.; Ozin, G. A. *Nat. Catal.* **2019**, *2*, (10), 889-898.
- [2] L. Wang, Y. Dong, T. Yan, Z. Hu, A. A. Jelle, D. M. Meira, P. N. Duchesne, J. Y. Y. Loh, C. Qiu, E. E. Storey, Y. Xu, W. Sun, M. Ghossoub, N. P. Kherani, A. S. Helmy, G. A. Ozin, *Nat. Commun.* **2020**, *11*, 2432.
- [3] M. Li, P. Li, K. Chang, T. Wang, L. Liu, Q. Kang, S. Ouyang, J. Ye, *Chem Commun* **2015**, *51*, 7645-7648.
- [4] J. Jia, H. Wang, Z. Lu, P. G. O'Brien, M. Ghossoub, P. Duchesne, Z. Zheng, P. Li, Q. Qiao, L. Wang, A. Gu, A. A. Jelle, Y. Dong, Q. Wang, K. K. Ghuman, T. Wood, C. Qian, Y. Shao, C. Qiu, M. Ye, Y. Zhu, Z. H. Lu, P. Zhang, A. S. Helmy, C. V. Singh, N. P. Kherani, D. D. Perovic, G. A. Ozin, *Adv Sci (Weinh)* **2017**, *4*, 1700252.
- [5] Y. F. Li, W. Lu, K. Chen, P. Duchesne, A. Jelle, M. K. Xia, T. E. Wood, U. Ulmer, G. A. Ozin, *J. Am. Chem. Soc.* **2019**, *141*, 14991-14996
- [6] H. Zhang, T. Wang, J. Wang, H. Liu, T. D. Dao, M. Li, G. Liu, X. Meng, K. Chang, L. Shi, T. Nagao, J. Ye, *Adv. Mater.* **2016**, *28*, 3703-3710.
- [7] J. Zhao, Q. Yang, R. Shi, G. I. N. Waterhouse, X. Zhang, L.-Z. Wu, C.-H. Tung, T. Zhang, *NPG Asia Mater.* **2020**, *12*, 5.
- [8] Y. Li, J. Hao, H. Song, F. Zhang, X. Bai, X. Meng, H. Zhang, S. Wang, Y. Hu, J. Ye, *Nat. Commun.* **2019**, *10*, 2359.
- [9] F. Sastre, A. V. Puga, L. Liu, A. Corma, H. García, *J. Am. Chem. Soc.* **2014**, *136*, 6798-6801.

11. Page 7, top (flow reactor measurements): The flow reactor measurements were conducted over the high-activity Ni₁₂P₅/SiO₂ catalyst at 290 C, yet much lower activities were measured than in the batch reactor measurements. If the reviewer is interpreting correctly, the activities measured in the

batch reactor at 290 C under light illumination are nearly two orders of magnitude lower than in the batch reactor study. Why is this, and how do the flow reactor activities compare with relevant data in the literature. These results highlight how photocatalytic activities reported in the literature tend to be very dependent on the reaction conditions used, making comparisons difficult. The authors should comment on this.

Re: Thanks again for the reviewer's comments. The setup of these two reactors is quite different. Firstly, the light intensity used in the batch reactor is 2.3 W cm^{-2} , while this value is 0.8 W cm^{-2} in the flow reactor. Moreover, in the batch reactor a flat microfiber filter was used to support the catalyst, while in the flow reactor the catalyst particles were packed into a tubular quartz capillary. It is conceivable that the catalyst in the batch reactor is more efficiently exposed to the light irradiation compared to in the flow reactor, where the incident light penetrates less into the catalyst in the tubular capillary. As a result, the local temperature of the catalyst surface in the batch reactor was higher than that in the flow reactor, resulting in a better catalytic performance.

As summarized in Table R1, despite the variations among the reactor setups, under similar light irradiation conditions the Ni_{12}P_5 and $\text{Ni}_{12}\text{P}_5/\text{SiO}_2$ display superior photothermal CO_2 conversion performance compared to the mentioned metal oxides and noble metal catalysts, demonstrating the advantages of the nickel phosphide photothermal catalyst.

12. Page 7, sintering of Ni/SiO_2 : The dramatic sintering of the Ni/SiO_2 catalyst seems surprising under the reaction conditions used. The authors should report the temperature of the catalyst during the measurements. The sintering might be due to the formation of $\text{Ni}(\text{CO})_4$ under reaction conditions, thus facilitating transport of Ni for particle growth. Transmission IR spectra of CO on Ni/SiO_2 , $\text{Ni}_{12}\text{P}_5/\text{SiO}_2$ and $\text{Ni}_2\text{P}/\text{SiO}_2$ catalysts have been reported previously (e.g., Journal of Catalysis 231 (2005) 300–313) and it was shown that $\text{Ni}(\text{CO})_4$ formed on Ni/SiO_2 , but the formation of this species was suppressed on the Ni phosphide catalysts. Another relevant IR study is the following: J. Phys. Chem. B 2004, 108, 10930-10941.

Re: Thanks for reviewer's helpful information. The test on metallic Ni was conducted at an apparent temperature of 290°C under illumination. However, because metallic Ni has its own photothermal effect, the local temperature should be higher than 290°C .

It's likely the $\text{Ni}(\text{CO})_4$ can be formed from Ni and CO as low as 50°C , and would decompose back to nickel and carbon monoxide at temperature about $220\text{-}250^\circ\text{C}$.¹ Historically this is the famous Mond process used to extract and purify nickel. According to the listed references, the $\text{Ni}(\text{CO})_4$ species can be detected by FTIR spectroscopy when using the metallic nickel as the CO_2 hydrogenation catalyst. Therefore, in our reaction condition, the formation and subsequent decomposition of $\text{Ni}(\text{CO})_4$ would cause serious sintering of Ni. However, the formation of $\text{Ni}(\text{CO})_4$ species on nickel phosphide was suppressed, which may contribute to its high stability with respect to particle size and catalytic performance. This point has been clarified in the revised manuscript with the related literatures being cited.

References:

[1] L. Mond, C. Langer, F. Quincke, *J. chem. Soc. Trans.* **1890**, 57, 749-753.

13. Figure 5a: The IR peak for terminally-bonded CO noticeably broadens as the Ni₁₂P₅ loading decreases. Not surprisingly, this suggests that there are a range of CO bonding environments on the Ni₁₂P₅ particles and that this becomes more pronounced for smaller particles. The authors should discuss the relevancy of this to the proposal of Ni₁₂P₅ providing “atom-precise” catalytic sites.

Re: Thanks for reviewer’s careful observation. As shown below (Figure R3), we think it is the peak height difference that make the “terminally-bonded CO” peak seems broadened. In the DRIFT test the samples were compacted into a small reaction chamber of a Harrick cell. Therefore, a higher loaded Ni₁₂P₅/SiO₂ sample would have more sites for CO bonding, which could be responsible for the higher peak height and apparent narrower line width.

Nevertheless, the reviewer is right that the precise CO bonding environments may change somewhat as the particle size shrinks. Based on our experimental findings and the aforementioned discussion of the invariance of the atom-precise nickel nanocluster superlattice to dispersity this point is clarified in the revised version of the manuscript

Figure R3. Expanded view of the terminally-bonded vCO stretching mode of Fig. 5a

14. Page 8, top (IR spectral interpretation): The authors should cite relevant literature that have assigned and interpreted the IR spectra of adsorbed CO on Ni phosphide catalysts.

Re: Thanks for reviewer’s advice. In the revised version the following references are included:

[1] Layman, K. A.; Bussell, M. E. *The J. Phys. Chem. B* **2004**, 108, 10930-10941.

[2] Sawhill, S. J.; Layman, K. A.; Van Wyk, D. R.; Engelhard, M. H.; Wang, C.; Bussell, M. E. *J. Catal.* **2005**, 231, 300-313.

15. Page 8, middle (Compared to metallic Ni,...): It is suggested that the selectivity of metallic Ni or single-atom Ni catalysts “would change a lot” under the reaction conditions probed – the authors should be specific in making this statement.

Re: Usually heterogeneous CO₂ hydrogenation over nickel catalysts involve two competing reactions:

the reverse water gas shift reaction for CO production ($\text{H}_2 + \text{CO}_2 \rightarrow \text{CO} + \text{H}_2\text{O}$), and the Sabatier reaction ($4\text{H}_2 + \text{CO}_2 \rightarrow \text{CH}_4 + 2\text{H}_2\text{O}$) for CH_4 production. Many studies have demonstrated that the initial CO_2/H_2 ratio, CO_2 conversion, temperature, catalyst particle size can influence the CO_2 hydrogenation selectivity. For example, it has been reported that a higher proportion of H_2 in the feed usually favors a high selectivity towards CH_4 production over.¹ In addition, increasing the conversion of CO_2 or raising the reaction temperature would also improve the selectivity to CH_4 on nickel catalyst.^{1,2} The nickel in the catalyst also plays an important role in the reaction selectivity, where a smaller particle size would favor the formation of CO rather than CH_4 .^{2,3} The size effect was more pronounced in the case of a Ni single atom catalyst, on which the CH_4 production was almost eliminated.⁴ However, at high temperature ($>350^\circ\text{C}$), single atom Ni would aggregate into Ni clusters of about 10 nm, resulting a drop of activity and selectivity.

In our study, we observed that factors such as the reaction temperature, particle size, CO_2/H_2 ratio and CO_2 conversion do not influence the CO_2 hydrogenation reaction pathways (always near 100% toward RWGS reactions), which distinguishes the Ni_{12}P_5 from other Ni based catalysts for all the reasons discussed above.

These clarifications have been added in the revised manuscript.

References:

- [1] G. A. Du, S. Lim, Y. H. Yang, C. Wang, L. Pfefferle, G. L. Haller, *J. Catal.* **2007**, *249*, 370-379.
- [2] H. C. Wu, Y. C. Chang, J. H. Wu, J. H. Lin, I. K. Lin, C. S. Chen, *Catal. Sci. Technol.* **2015**, *5*, 4154-4163.
- [3] C. Vogt, E. Groeneveld, G. Kamsma, M. Nachtegaal, L. Lu, C. J. Kiely, P. H. Berben, F. Meirer, B. M. Weckhuysen, *Nat. Catal.* **2018**, *1*, 127-134.
- [4] M. M. Millet, G. Algara-Siller, S. Wrabetz, A. Mazheika, F. Girgsdies, D. Teschner, F. Seitz, A. Tarasov, S. V. Leychenko, R. Schlogl, E. Frei, *J. Am. Chem. Soc.* **2019**, *141*, 2451-2461.

16. Page 8, bottom: For the Ni_{12}P_5 particle range studied (7.8-13.3 nm), the distribution of exposed facets doesn't change much with particle size as the biggest changes happen for particle diameters below ~5 nm.

Re: According to the size distribution statistics, the smallest mean particle size was obtained on the lowest loading sample (3.1 wt% $\text{Ni}_{12}\text{P}_5/\text{SiO}_2$), which is 7.8 nm and more than one third of the particles possessed a size of below 5.5 nm.

One of the unique findings is that the larger Ni_{12}P_5 particles still favor the CO production pathway rather than CH_4 , distinct to metallic nickel catalysts. As mentioned, the increased Ni-Ni bond length in Ni_{12}P_5 and Ni-P ensemble effects, which are invariant to the particle size, could be responsible

17. Page 9, top: The sentence ending with "such as employing it in tandem with other catalysts known for their capacity for CO conversion reactions" is unclear to the reviewer. The authors should explain this more fully.

Re: It is true that research on tandem catalysts have received a lot of attentions recently.¹ Take olefin production for example. Beside the direct conversion of CO₂ into olefins over a single catalyst, the reaction pathways can be divided into two sequential steps, such as the RWGS reaction for CO production followed by CO-mediated Fischer–Tropsch synthesis to olefins. In each reaction different specialized catalysts would be adopted, constructing the so-called “tandem catalyst”. One of the merits of the tandem catalyst over a single component catalyst is the ease with which to control the reaction pathway, selectivity, and catalyst stability. In our work, we have discovered that the Ni₁₂P₅ is able to act as a selective RWGS catalyst with high activity and stability. As mentioned, the near-unity selectivity for CO generation can be well retained as the reaction conditions change, potentially an asset if integrated into a tandem catalyst.

References:

[1] Z. Q. Ma, M. D. Porosoff, *ACS Catal.* **2019**, *9*, 2639-2656.

18. Page 9, top: The sentence “This atom - perfect design philosophy is expected to be applicable to other compounds...” is quite strong and needs to be considered in light of the comments above, and more evidence for atom-perfect design provided if it is to be included in the manuscript.

Re: This is a fair point and we have expanded upon our thinking in the revised manuscript. In our study the Ni₁₂P₅ and CoP nanomaterials showed high photothermal RWGS activity with near-unity selectivity, which has been extended to other nickel phosphides with different Ni/P ratios as well as other metal phosphide materials not reported herein.

Reviewer #2 (Remarks to the Author):

This work describes the use of Ni₁₂P₅ and its supported form as an active photothermal catalyst for RWGS. The manuscript is clearly written and their good activity and high stability have been highlighted. The materials are also well characterized. After going through the details; however, I evaluate that the novelties and impacts of the contents are not sufficiently high for the journal under evaluation. Therefore, I do not recommend this manuscript for publication in Nature Communications. Some detailed comments are listed below.

Re: We thank the reviewer for these helpful comments.

- The term “photocatalysis” in the title is somewhat misleading. Probably much more precise to call it “photothermal catalysis”

Re: We agree with the reviewer’s point. In the revised manuscript the word “photocatalysis” was changed to “photothermal catalysis”

- What is the reason for the decreased absorbance of Ni₁₂P₅/SiO₂ at longer wavelength? Can the UV-Vis spectrum of SiO₂ included in the SI?

Re: According to the reviewer's comments, we have tested the UV-Vis-NIR spectra of Ni_{12}P_5 , $\text{Ni}_{12}\text{P}_5/\text{SiO}_2$ and SiO_2 (shown in Figure R4), and the corresponding plots have been added into the revised version as Fig. 2b.

As shown below, both the Ni_{12}P_5 and $\text{Ni}_{12}\text{P}_5/\text{SiO}_2$ exhibited broadband absorption throughout UV-Vis-NIR region. The absorption in the NIR is less than the UV-Vis region, and this phenomenon is more pronounced on $\text{Ni}_{12}\text{P}_5/\text{SiO}_2$ samples. This could be related to the poorer Mie scattering ability of the smaller particle size loaded Ni_{12}P_5 sample. Also note, SiO_2 is almost transparent through the whole UV-VIS-NIR region, which indicates that it would not contribute to the optical absorption of the $\text{Ni}_{12}\text{P}_5/\text{SiO}_2$ samples.

Figure R4. UV-vis-NIR spectra of Ni_{12}P_5 , SiO_2 and $\text{Ni}_{12}\text{P}_5\text{-SiO}_2$ samples being loaded onto a binder-free borosilicate glass microfiber filter.

- Regarding the TEM study, is it reasonable and justified to report so accurate numbers with the significant figures (e.g. 86.1 ± 30.2 nm and 0.604 nm and 0.401 nm)? The same holds for other values (e.g. EXAFS: 0.163 nm and 0.264 nm).

Re: The particle size, lattice spacing, EXAFS values are read from the software in data processing. In the revised manuscript we have corrected the number of significant figures. For example, the previous " 86.1 ± 30.2 " was changed as " 86 ± 30 "

- Similarly, selectivity is shown up to 1 decimal point. Can it be so accurate? Standard deviation?

Re: As mentioned above, in the revised manuscript we have reconsidered the selection of significant figures, and the standard deviations are shown in the manuscript.

- What are other products observed? Methane?

Re: As show below (Figure R5), the GC spectra performed on both batch and flow reactors indicate other than CO, the only byproduct is a tiny amount of methane. Other products such as paraffin/olefin and methanol were not observed.

Figure R5. The GC traces from testing in **a**, batch reactor and **b**, flow reactor. The insets are corresponding expanded view.

- Fig. 1 is beautiful, but I do not see much the relevance in this work.

Re: This point has been clarified in the responses to referee 1 to which this referee is referred. In our revised manuscript, Fig. 1 along with the statements are modified, please see the yellow-marked words in the introduction part.

- Why were 3 psi H₂ and 15 psi CO₂ and the specific ratio chosen? Pressure unit should be changed.

Re: The initial total pressure in the batch reactor was chosen as about 18 psi to simulate the ambient pressure (14.7 psi). After the pressure unit conversion this value is 1.2 bar. In the revised manuscript, we have noted the pressure as both psi and bar unit.

The initial CO₂/H₂ ratio of 5:1 was selected for the following reasons. As shown below (Figure R6), in a batch reactor where the Xe lamp was the only energy source and no external heating was provided, increasing the CO₂/H₂ ratio from 1:1 to 5:1, the CO production rate drastically improved, while further increase to 7.5:1 finally lead to a drop of the production rate. In addition, the selectivity to the RWGS reaction and CO production would not change with the CO₂/H₂ ratio, which was always above 99.5%.

One plausible explanation for the better CO₂ hydrogenation performance in a CO₂ rich atmosphere was that the low thermal conductivity of the CO₂ gas may lead to a higher local temperature on the catalyst surface. According to the literature, the thermal conductivity of H₂ is about 10-times higher than that of CO₂ (186.9 mW/m k vs 16.8 mW/m k). The high thermal conductivity of the H₂ gas indicates that it can enhance the heat transfer between the hot catalyst surface and the gaseous atmosphere. Since the tests in the batch reactor were performed just under light illumination without any external heating, a higher proportion of H₂ would result in a significant heat loss and thereby lowering the local temperature on the catalyst surface along with the RWGS reaction rate. Nevertheless, further decreasing the H₂ proportion would be deleterious to the reaction rate as the H₂ itself is a reactant.

Overall, these results offered a perspective of the effect of gas composition on the photothermal

performance.

Figure R6. The normalized CO production rate and selectivity plots from batch reactor testing under various CO₂/H₂ ratio atmospheres. The initial total pressure was controlled as 18 psi (1.2 bar). The tests were performed under 2.3 W cm⁻² illumination without external heating.

- Conversion of gases and equilibrium conversions should be explicitly mentioned.

Re: Thanks for reviewer's comment. We have added the time course plots (Figure R7) to explicitly display the of CO₂ conversion and reaction equilibrium point. Please see the Supplementary Fig. 9 in the revised version.

- Regarding the estimation of the catalyst temperature, the equilibrium conversion at different temperatures was used. This is only possible when the conversion is very high and at the equilibrium (according to the text, it is the case). It would be important to show the temporal evolution of the reaction performance for one or a few specific case(s) so that one can indeed reach the equilibrium conversion.

Re: Thanks for reviewer's suggestion. As shown in Figure R7, conversion of CO₂ as a function of time was plotted. Accordingly, the CO₂ conversion remained unchanged after 5 h of illumination. To guarantee the reaction has reached equilibrium we further extended the reaction time to 11.5 h, at which point the CO₂ conversion data was subsequently used to estimate the local temperature.

Figure R7. Time course CO₂ conversion plots

- Catalytic performance under thermal activation should be reported.

Re: According to the reviewer's comment, we have presented the catalytic performance over the best sample (10.4 wt% Ni₁₂P₅/SiO₂) in the supplementary information. As shown in Figure R8, the test was conducted in a flow reactor, where a temperature controller along with electric heater were implemented to control the temperature. Under thermal activation (dark condition), the 10.4 wt% Ni₁₂P₅/SiO₂ sample can afford a CO production rate of 8.03 mmol g_{cat}⁻¹ h⁻¹ at 320°C. As a comparison, the CO production rate under light conditions (i.e. both thermal and photo activation) at the same apparent temperature was 25.57 mmol g_{cat}⁻¹ h⁻¹, corresponding to a 2.2-fold enhancement. This enhancement factor was even more significant at a lower apparent temperature, indicating the advantages of the photothermal effect of the Ni₁₂P₅ catalyst.

The related discussions and charts have been added in the revised manuscript, please see the Supplementary Figure 11 and yellow-marked words in Page 8.

Figure R8. Catalytic performance of 10.4 wt% Ni₁₂P₅/SiO₂. The tests were performed in a flow reactor at different reaction temperatures with and without solar irradiation. The gas flow contained 2.5 sccm of CO₂ and 0.5 sccm of H₂.

- Regarding the catalytic tests performed under continuous flow (Figs. 4c and 5d), it says "controlled apparent temperature of 290 °C" for the former. This means that the sample heated? Similar to above, what is the catalytic activity without illumination?

Re: Yes, the sample was heated when tested in a flow reactor. As shown above, in dark conditions without illumination, as the temperature increased from 160 °C to 320 °C the CO production rate gradually increased, which is in accordance with the endothermicity of the RWGS reaction. It is worth noting that under light illumination, the CO production rate can be significantly improved, demonstrating the solar advantage of the photothermal effect for the Ni₁₂P₅ catalyst.

- The details of the setup (light source, focus, etc.) should be described ideally with photograph of

the setup.

Re: Pictures of the light source and reactors are shown below (Figure R9 and Figure R10). They have also been added into the revised supplementary information, Fig. 17-18.

Figure R9. Digital photograph of the batch reactor setup.

Figure R10. Digital photograph of the flow reactor setup.

Reviewer #3 (Remarks to the Author):

The authors presented a novel application of metal phosphide based catalyst for photocatalytic RWGS applications. This is an important piece of work which would help to boost the study of transitional metal phosphide in critical industrial processes. However, several issues need to be rectified before the manuscript is ready for publications.

Re: Thanks for reviewer's positive and helpful comments.

1. It is unclear from the work if the RWGS by Ni_{12}P_5 is photocatalytic or simply a photothermal effect.

Re: Based on our catalytic tests run at different wavelengths it seems most likely the Ni₁₂P₅ is a photothermal catalyst.

2. Is this a promising approach to achieve sustainable RWGS reaction? What is the performance of the supported Ni₁₂P₅ catalyst compared to other reported RWGS catalysts?

Re: As shown in the table and mentioned in the response to referee 1, the photothermal CO₂ hydrogenation performances of some reported catalysts show that Ni₁₂P₅ and Ni₁₂P₅/SiO₂ have a superior CO₂ conversion rate compared to representative metal oxides and noble metal catalysts using light activation without any external heating, demonstrating its potential for a sustainable RWGS reaction. Table R2 was added into the revised supplementary information.

Table R2. Comparison of CO₂ conversion rates for some CO₂ hydrogenation catalysts under light irradiation without external heating.

Catalyst	Light source	Feed composition	CO ₂ conversion rate	Selectivity	
				CO	CH ₄
Cu ₂ O ¹	300 W Xe light (full spectrum, 40 suns)	CO ₂ /H ₂ =83/17	70.3 mmol g ⁻¹ h ⁻¹	~100	
In ₂ O _{3-x} ²	300 W Xe light (~20 suns)	CO ₂ /H ₂ =50/50	238.8 mmol g ⁻¹ h ⁻¹	~100	
Pt/NaTaO ₃ ³	300 W UV-enhanced Xe lamp	CO ₂ /H ₂ =50/50	140.5 μmol g ⁻¹ h ⁻¹	99	1
Pd@Nb ₂ O ₅ ⁴	300 W Xe lamp	CO ₂ /H ₂ =50/50	1.8 mmol g ⁻¹ h ⁻¹	100	
Cu/Pd/HyWO _{3-x} ⁵	300 W Xe lamp (1 W cm ⁻²)	CO ₂ /H ₂ =50/50	40.8 μmol g ⁻¹ h ⁻¹	100	
Fe@C ⁶	300 W Xe lamp	CO ₂ /H ₂ =50/50	26.1 mmol g ⁻¹ h ⁻¹	100	
FeO-CeO ₂ ⁷	300 W Xe lamp (2.2 W cm ⁻²)	CO ₂ /H ₂ /Ar=15/60/25	20 mmol g ⁻¹ h ⁻¹	97-99.9%	
SA Ni/Y ₂ O ₃ ⁸	ambient daytime sunlight (from 0.52 to 0.7 kW m ⁻²)	CO ₂ /H ₂ /N ₂ =2.5/10/87.5	7.5 L m ⁻² h ⁻¹		100
Ni/SiO ₂ ·Al ₂ O ₃ ⁹	solar simulator	CO ₂ /H ₂ /N ₂ =15/70/15	14.4 mmol g ⁻¹ h ⁻¹	2.8	97.2
NiO ⁹	solar simulator	CO ₂ /H ₂ /N ₂ =15/70/15	13.3 mmol g ⁻¹ h ⁻¹		100
Ni ₁₂ P ₅ (this work)	300 W Xe lamp (2.3 W cm ⁻²)	CO ₂ /H ₂ =83/17	155.7 mmol g ⁻¹ h ⁻¹	99.5	0.5
Ni ₁₂ P ₅ /SiO ₂ (this work)	300 W Xe lamp (2.3 W cm ⁻²)	CO ₂ /H ₂ =83/17	960.3 mmol g ⁻¹ h ⁻¹	99.7	0.3

Co ₂ P (this work)	300 W Xe lamp (2.3 W cm ⁻²)	CO ₂ /H ₂ =83/17	15.7 mmol g ⁻¹ h ⁻¹	99.2	0.8
Co ₂ P/SiO ₂ (this work)	300 W Xe lamp (2.3 W cm ⁻²)	CO ₂ /H ₂ =83/17	227.7 mmol g ⁻¹ h ⁻¹	99.5	0.5

References:

- [1] Wan, L. L.; Zhou, Q. X.; Wang, X.; Wood, T. E.; Wang, L.; Duchesne, P. N.; Guo, J. L.; Yan, X. L.; Xia, M. K.; Lie, Y. F.; Jelle, A. A.; Ulmer, U.; Jia, J.; Li, T.; Sun, W.; Ozin, G. A. *Nat. Catal.* **2019**, *2*, (10), 889-898.
- [2] L. Wang, Y. Dong, T. Yan, Z. Hu, A. A. Jelle, D. M. Meira, P. N. Duchesne, J. Y. Y. Loh, C. Qiu, E. E. Storey, Y. Xu, W. Sun, M. Ghossoub, N. P. Kherani, A. S. Helmy, G. A. Ozin, *Nat. Commun.* **2020**, *11*, 2432.
- [3] M. Li, P. Li, K. Chang, T. Wang, L. Liu, Q. Kang, S. Ouyang, J. Ye, *Chem. Commun.* **2015**, *51*, 7645-7648.
- [4] J. Jia, H. Wang, Z. Lu, P. G. O'Brien, M. Ghossoub, P. Duchesne, Z. Zheng, P. Li, Q. Qiao, L. Wang, A. Gu, A. A. Jelle, Y. Dong, Q. Wang, K. K. Ghuman, T. Wood, C. Qian, Y. Shao, C. Qiu, M. Ye, Y. Zhu, Z. H. Lu, P. Zhang, A. S. Helmy, C. V. Singh, N. P. Kherani, D. D. Perovic, G. A. Ozin, *Adv. Sci.* **2017**, *4*, 1700252.
- [5] Y. F. Li, W. Lu, K. Chen, P. Duchesne, A. Jelle, M. K. Xia, T. E. Wood, U. Ulmer, G. A. Ozin, *J. Am. Chem. Soc.* **2019**, *141*, 14991-14996
- [6] H. Zhang, T. Wang, J. Wang, H. Liu, T. D. Dao, M. Li, G. Liu, X. Meng, K. Chang, L. Shi, T. Nagao, J. Ye, *Adv. Mater.* **2016**, *28*, 3703-3710.
- [7] J. Zhao, Q. Yang, R. Shi, G. I. N. Waterhouse, X. Zhang, L.-Z. Wu, C.-H. Tung, T. Zhang, *NPG Asia Mater.* **2020**, *12*, 5.
- [8] Y. Li, J. Hao, H. Song, F. Zhang, X. Bai, X. Meng, H. Zhang, S. Wang, Y. Hu, J. Ye, *Nat. Commun.* **2019**, *10*, 2359.
- [9] F. Sastre, A. V. Puga, L. Liu, A. Corma, H. García, *J. Am. Chem. Soc.* **2014**, *136*, 6798-6801.

3. I find it hard to discern from Figure 3c if there is any change to the presented in-situ XANES spectra.

Re: In the time-resolved in-situ XANES test, no significant changes were observed in the XANES spectra. As shown below, the expanded view of the XANES plots indicate the variations in intensity did not show any obvious changes or trends pointing out the high stability and performance of the Ni₁₂P₅ material for the RWGS reaction.

Figure R11. Expanded view of the XANES plots at different times-on-stream. The test was performed under reaction conditions $\text{CO}_2:\text{H}_2=5:1$ gas flow, with a temperature of 300 °C.

4. Why was a lower light intensity used for the long term stability testing in a flow reactor? From the results presented, the performance of the catalyst was reduced by over 100 times. Is the stability of the catalyst reproducible under the same illumination intensity as the batch reaction?

Re: Due to the different catalyst test systems used in our work, the light intensity used in a flow reactor is lower than that in a batch reactor. Long-term testing under higher light intensity would be interesting for future studies in a scaled demonstrator.

5. The authors state that the observed performance can be attributed in the periodic array of uniform and atom - precise Ni nanoclusters in Ni_{12}P_5 , a nanocluster superlattice catalyst, which disfavors strong multi - coordinate bonding to CO. However, the nature of the Ni_{12}P_5 active site(s) during the catalytic reaction is not clear. Is TEM alone sufficient to support the retainment of the Ni_{12}P_5 structure after the catalytic reaction?

Re: In this work, maintenance of the structure of the Ni_{12}P_5 catalyst was confirmed by various characterization methods. First, no obvious changes were observed in XPS and XRD studies of the spent catalyst, indicating the surface and bulk structure-properties of the Ni_{12}P_5 catalyst were well-preserved after catalytic testing. Moreover, time-resolved in situ XANES under flow reaction conditions $\text{CO}_2:\text{H}_2=5:1$, with a temperature of 300 °C or 350 °C, further confirmed retention of the high stability of Ni_{12}P_5 during the catalytic process. Considering the constant CO_2 hydrogenation reactivity and selectivity over the Ni_{12}P_5 catalyst during a 100-h test, we assume the structure of Ni_{12}P_5 is well-sustained after the catalytic reaction.

6. Corresponding UV-Vis-NIR spectra would be helpful to support the authors' attribution of the decreasing TOF with lower Ni_{12}P_5 loading to poorer light harvesting ability. Nonetheless, a performance test at the same temperature in the absence of light would be a better experiment to prove the authors' idea.

Re: Thanks for the reviewer's suggestion. In the revised manuscript we have presented the UV-Vis-

NIR absorption spectra of all the samples. As shown below, as the loading amount decreased, the A% was also reduced.

Figure R12. UV-vis-NIR spectra of Ni_{12}P_5 , SiO_2 and $\text{Ni}_{12}\text{P}_5\text{-SiO}_2$ samples loaded onto a binder-free borosilicate glass microfiber filter support (0.5 mg/cm^2).

7. Additional comments:

a. The following sentence is too convoluted:

Combined with the aforementioned photocatalytic tests performed on Ni_{12}P_5 with varying particle diameters, these observations indicate that the nearly 100% CO selectivity of the Ni_{12}P_5 catalyst was virtually independent of overall particle size, the gas composition and the reaction temperature, in other words, beyond the prevailing view of the size effect.

Re: In the revised manuscript, this sentence has been modified to the following:

Combined with the aforementioned photocatalytic tests performed on Ni_{12}P_5 with various particle sizes, these observations indicate that the near 100% CO selectivity for the Ni_{12}P_5 catalyst was virtually independent of overall particle size, gas composition and reaction temperature, a distinctive feature of this class of nickel catalyst.

b. The discussion section only consists of a summary of the work.

Re: Thanks for the reviewer's comment. In the revised manuscript, the structure and content of the discussion section has been modified.

c. Poor selection of significant figures, for example: $960.3 \pm 11.7 \text{ mmol g}_{\text{cat}}^{-1} \text{ h}^{-1}$

Re: Thanks for the reviewer's reminder. We have reduced the number of significant figures from $960.3 \pm 11.7 \text{ mmol g}_{\text{cat}}^{-1} \text{ h}^{-1}$ was adjusted as 960 ± 12 now.

d. Missing punctuation:

The positions of the aforementioned two peaks are the same, regardless of the loading and size of the Ni₁₂P₅ particles. The intensity of the former peak overwhelms that of the latter, indicating that most CO molecules preferred bonding individually to separated Ni atoms over engaging in multi-coordinate bonding.

Re: Thanks for all reviewer's careful reading which has helped us improve the quality of the manuscript.

REVIEWER COMMENTS

Reviewer #1 (Remarks to the Author):

The authors have made significant revisions to the manuscript and it is most definitely improved. The findings for photothermal RWGS conversion of CO₂ to CO with near 100% selectivity over Ni₁₂P₅/SiO₂ are quite impressive, but the reviewer continues to have some substantive concerns, which are noted below.

1. Page 1 and Figure 1: The reviewer continues to find the figure and associated text to be potentially misleading. The suggestion seems to be that the Ni₁₂P₅/SiO₂ catalysts are most closely represented by structure d, when in reality structure b is most representative based on the physical evidence presented – TEM images (Figure S2) and particle size histograms (Figure S3). The particle size distributions are quite broad, as noted in the previous review. Readers will use this figure to form a concept of the structure of the catalyst - Ni₁₂P₅ particles dispersed on the SiO₂, which is separate from the concept of a Ni cluster superlattice. The Ni₁₂P₅ particles have high polydispersity and the TEM certainly points to them not being uniformly distributed on the silica support. Also, a more rigorous case needs to be made for an atomically precise Ni site within a Ni cluster, and this relates to the solid-state and surface properties of Ni₁₂P₅ (and should be considered separately from the support). The authors have not addressed these points in a convincing fashion in the introduction and in the analysis and discussion of the relevant findings.

2. The reviewer concerns in comment #1 relate to the evidence the authors present for the Ni₁₂P₅/SiO₂ catalysts being “a periodic array of atom precise Ni clusters”. The concept is appealing, and this could be a further boast to prominence of metal phosphides being unique and versatile catalysts for conventional, photo and photothermal catalytic processes. However, the bar of evidence needs to be high here with hard evidence to support the prominence that comes with publication in Nature Communications. The primary evidence presented by the authors is near unity selectivity towards the RWGS product (CO) and IR spectra for adsorbed CO indicating only linearly bonded CO. But examination of Table S2 indicates a fairly broad range of catalysts have near 100% selectivity for the photocatalytic conversion of CO₂ to CO including Cu₂O, In₂O_{3-x}, noble metals (Pt, Pd) on supports, and Fe and FeO on supports. Does the >99% CO selectivity of these catalysts qualify them as atom precise catalysts? What qualifies and disqualifies a nanoparticle catalyst as being atom precise? Single crystalline nanoparticles? Resistance to sintering?

3. Table 1 and accompanying text – Ni dispersions and turnover frequencies: How were the Ni dispersions and turnover frequencies calculated? The details of these calculations (including equations used) and the data used should be included in the Supporting Information. The decrease in TOF with Ni₁₂P₅ particle size needs to be carefully explained and should be supported by quantitative evidence than stating that it is due to reduced light absorption. In this regard, it would be helpful and perhaps more persuasive if the authors would include quantum efficiencies for each catalyst. Shouldn't the TOF of an atom precise catalyst be independent of particle size since all sites are equivalent?

4. Figure 5a: This figure should be replaced by Figure R3 from the author rebuttal document since it shows a narrower range of wavenumbers. Also, it would be helpful if the Y axis was expanded so that the IR peaks are larger. The reviewer is not convinced by the author rebuttal that there is no peak broadening as the Ni₁₂P₅ loading is lowered. The authors should calculate and report FWHM for each spectrum to support their argument that there is no peak broadening. Also, there is pronounced asymmetry of the IR peaks that suggests a range of sites for linear bonded CO species. The authors should comment on this point.

5. Figure 5b-d: The surface crystal structures correctly show that there are two distinct types of Ni sites in Ni₁₂P₅? Might these distinct Ni sites have different interactions with CO₂ and CO? This could explain the peak broadening of the IR spectra. Oyama has proposed that the distribution of

tetrahedral Ni(1) and square pyramidal Ni(2) sites at the surface of Ni₂P particles changes with particle size (based on EXAFS data). This could be relevant for Ni₁₂P₅ also. A recently published DFT study (Energy Technology 2019, 7, 1900013; <https://doi.org/10.1002/ente.201900013>) examines the low Miller index surfaces of Ni₁₂P₅ with a focus on determining potential active sites (for HER). The results of this study should be considered and discussed in the context of the proposed atomically precise Ni clusters being the active sites for the photothermal RWGS reaction.

Reviewer #2 (Remarks to the Author):

The authors revised and improved the manuscript taking most comments of the reviewers into account. The great efforts are highly appreciated. After going through all answers and actions taken and carefully reading again the whole manuscript, I conclude that the reliable novelties and the details are still unclear, despite that the contents are well presented in a way that the messages are well highlighted. Therefore, I still do not think the quality of the manuscript meets the standard of the journal under consideration. A few further remarks are given below.

- As commented also by the other reviewer, the origin of the catalytic activity is not clear. If it is by photothermal effects, the thermal catalytic activity should be more systematically studied. Probably checking other more standard materials in the reactor system would improve the reliability of this work.
- Higher catalytic activity with illumination is reported, but this can be simply the effect of heating by the energy of the lamp. The "photothermal" effect needs to be carefully studied. Can the catalyst temperature be measured?
- The rationale behind the choice of the material should be clearly given. According to Supplementary Table 2, there does not seem a consistency for a choice of the materials in literature. An explanation is expected.
- The precise and defined nature of the active sites are claimed. Even a single site nature and surface structure are presented and discussed, although their relationships with catalytic activity are not established. Elucidating this point is likely necessary to be impactful for the readers of the journal under consideration. I find Figure 1 is still misleading and confusing.
- The chromatograms shown in Supplementary Fig. 5 show that there are issues with gas separation and consequently the quantification.
- The catalytic activity reported in this study is much higher than those presented in literature (Supplementary Table 2). Based on the values given in this work, I quickly calculated the maximum amount of CO produced (i.e. at 100% H₂ conversion) and I obtained about 200 mmol g_{cat}⁻¹ (no h⁻¹ scaling – it will be much lower in practice by h⁻¹ scaling) as the maximum value (in reality I guess 1 order of magnitude less). In this work about 1000 mmol g_{cat}⁻¹ h⁻¹ is reported (i.e. 28 gCO g_{cat}⁻¹ h⁻¹ which is impossibly high under any condition using any catalyst). I guess the values must be checked carefully.

Reviewer #3 (Remarks to the Author):

I appreciate the efforts by the authors in response to the feedbacks by the reviewers. However, I found that not all responses by the authors are substantiated or reflected clearly in the manuscript.

1. I still find it premature in regards to the atomic perfection claim by the authors. Even the authors agreed that defects may be present which led to the formation of CH₄ during catalysis. In addition, faint shoulder at 1950 cm⁻¹ which corresponds to bridging CO can be observed in Figure R3.
2. The authors mentioned about the drop activity and selectivity temperature > 350 °C. However, no quantitative information was presented in Figure 5 or S11.
3. In response to reviewer 3, the authors stated that catalyst tests were performed at different

wavelengths, and suggest the photothermal nature of the catalyst based on the supposed results. However, it was not convincing as these observations were not discussed nor presented in the revised manuscript. I find it odd to exclude such direct observations from the manuscript. The clarification of the catalytic nature of the catalyst may be important with regards to the discrepancy between the batch and flow reactor.

Responses to Reviewers' Comments

Reviewer #1 (Remarks to the Author):

The authors have made significant revisions to the manuscript and it is most definitely improved. The findings for photothermal RWGS conversion of CO₂ to CO with near 100% selectivity over Ni₁₂P₅/SiO₂ are quite impressive, but the reviewer continues to have some substantive concerns, which are noted below.

Re: We gratefully appreciate the reviewer's recognition of our efforts and the most valuable comments that follow in his second review of our work that we will address to the best of our ability.

1. Page 1 and Figure 1: The reviewer continues to find the figure and associated text to be potentially misleading. The suggestion seems to be that the Ni₁₂P₅/SiO₂ catalysts are most closely represented by structure d, when in reality structure b is most representative based on the physical evidence presented – TEM images (Figure S2) and particle size histograms (Figure S3). The particle size distributions are quite broad, as noted in the previous review. Readers will use this figure to form a concept of the structure of the catalyst - Ni₁₂P₅ particles dispersed on the SiO₂, which is separate from the concept of a Ni cluster superlattice. The Ni₁₂P₅ particles have high polydispersity and the TEM certainly points to them not being uniformly distributed on the silica support. Also, a more rigorous case needs to be made for an atomically precise Ni site within a Ni cluster, and this relates to the solid-state and surface properties of Ni₁₂P₅ (and should be considered separately from the support). The authors have not addressed these points in a convincing fashion in the introduction and in the analysis and discussion of the relevant findings.

Re: After carefully considering the reviewer's comments, we realized the current Ni₁₂P₅ nanoparticles with a relatively broad size distribution and random facet exposure is not a suitable model to elaborate the "atomically precise". To this end, this concept has been removed and revised, based on the following considerations:

The key point we would like to emphasize is that the introduction of P into a metallic Ni lattice to form the Ni₁₂P₅ could significantly alter the catalytic nature of pristine metallic Ni counterparts. As shown in the manuscript, the linear bonded Ni-CO overwhelmed the bridge-bonded Ni-CO (although there is still a range of sites for linear bonded CO species, as the reviewer mentioned). This observation points to single-site reaction type behavior, explicable in terms of the expanded Ni-Ni bond length and the ensemble effect of P, which leads to well-separated few-atom Ni clusters. As a result, the Ni, a well-documented CO₂ methanation catalyst, now in the form of Ni₁₂P₅ turns out to be a CO producer with near unity selectivity. Moreover, such near unity selectivity was independent of the reaction conditions (e.g., temperature, CO₂/H₂ ratio) and particle size, which endows it with a distinctive feature among nickel-containing heterogeneous catalysts.

In the revised manuscript we have removed all mention of the “atomically precise” concept and recrafted the revised proposal throughout the abstract, introduction and discussion parts (highlighted in yellow).

2. The reviewer concerns in comment #1 relate to the evidence the authors present for the Ni₁₂P₅/SiO₂ catalysts being “a periodic array of atom precise Ni clusters”. The concept is appealing, and this could be a further boast to prominence of metal phosphides being unique and versatile catalysts for conventional, photo and photothermal catalytic processes. However, the bar of evidence needs to be high here with hard evidence to support the prominence that comes with publication in Nature Communications. The primary evidence presented by the authors is near unity selectivity towards the RWGS product (CO) and IR spectra for adsorbed CO indicating only linearly bonded CO. But examination of Table S2 indicates a fairly broad range of catalysts have near 100% selectivity for the photocatalytic conversion of CO₂ to CO including Cu₂O, In₂O_{3-x}, noble metals (Pt, Pd) on supports, and Fe and FeO on supports. Does the >99% CO selectivity of these catalysts qualify them as atom precise catalysts? What qualifies and disqualifies a nanoparticle catalyst as being atom precise? Single crystalline nanoparticles? Resistance to sintering?

Re: Most grateful for the reviewer’s comments. We have seriously considered the reviewer’s suggestion and made appropriate changes throughout as mentioned in the comment#1.

3. Table 1 and accompanying text – Ni dispersions and turnover frequencies: How were the Ni dispersions and turnover frequencies calculated? The details of these calculations (including equations used) and the data used should be included in the Supporting Information. The decrease in TOF with Ni₁₂P₅ particle size needs to be carefully explained and should be supported by quantitative evidence than stating that it is due to reduced light absorption. In this regard, it would be helpful and perhaps more persuasive if the authors would include quantum efficiencies for each catalyst. Shouldn’t the TOF of an atom precise catalyst be independent of particle size since all sites are equivalent?

Re: Most thankful for the reviewer’s valuable comment. We have re-calculated the TOF and details are offered as follows:

The TOF is calculated in terms of per nickel metal site, according to the following equation:

$$TOF(s^{-1}) = \frac{\text{Total CO turnovers per second}}{\text{Total Ni site numbers}}$$

Where the total CO turnovers per second is calculated as

$$\text{Total CO turnovers per second} = \frac{r \times \text{mass}_{\text{catalyst}}}{3600} = \frac{r \times \text{mass}_{\text{total}} \times x \times N_A}{3600}$$

Where r is the CO production rate ($\text{mmol g}_{\text{cat}}^{-1} \text{h}^{-1}$) presented in Table 1, $\text{mass}_{\text{total}}$ is the mass of the used catalyst (including Ni_{12}P_5 and SiO_2 supports), x is the weight percent of Ni_{12}P_5 in the $\text{Ni}_{12}\text{P}_5/\text{SiO}_2$, N_A is the Avogadro number.

To determine Ni site numbers, there are two methods:

Method 1: Estimate from metal dispersion result via H_2 adsorption measurement:

$$\text{Total Ni site numbers} = \frac{\text{mass}_{\text{total}} \times y \times D \times N_A}{M}$$

where y is the weight percent of Ni in the sample, as determined from ICP-OES, D is the metal dispersion which was obtained from a pulse H_2 chemisorption experiment. Note the metal dispersion herein is determined with respect to the Ni metal, and assume atomic hydrogen only binds to surface nickel atoms with a H:Ni stoichiometry of 1.¹⁻² M is the atomic weight of Ni, which is 58.69 g mol^{-1} .

Therefore, the TOF can be calculated according to the following equation:

$$\text{TOF}(\text{s}^{-1}) = \frac{r \times \text{mass}_{\text{total}} \times x \times N_A}{3600} \div \frac{\text{mass}_{\text{total}} \times y \times D \times N_A}{M} = \frac{r \times x \times M}{3600 \times y \times D}$$

The results are listed in the following table:

	Metal dispersion (%)	TOF (s^{-1})
Ni_{12}P_5	0.014	23.25
10.4 wt% $\text{Ni}_{12}\text{P}_5/\text{SiO}_2$	0.094	20.45
5.2 wt% $\text{Ni}_{12}\text{P}_5/\text{SiO}_2$	0.35	3.89
3.1 wt% $\text{Ni}_{12}\text{P}_5/\text{SiO}_2$	0.88	0.76

Method 2: Estimate from lattice structure

In this method the theoretical metal site concentration, L , was estimate based on the crystal structure of Ni_{12}P_5 , according to the paper published by Oyama et. al.³ The

theoretical metal site concentration assumes that the samples are composed of uniform spherical particles. This L is calculated by:

$$L = S \times n$$

where n is the average surface metal atom density, and S is the effective surface area.

The effective surface area was calculated as:

$$S = \frac{6}{\rho \times D}$$

where ρ is the material density (7.53 g cm^{-3} for Ni_{12}P_5), D is the average particle size extract from the size distribution statistics based on TEM images (as shown in Table 1 and Supplementary Fig. 3).

Figure R1. Crystal structure of Ni_{12}P_5 .

The average surface metal atom density is estimated via a published method.³ For the Ni_{12}P_5 there are six, six, and six Ni atoms on the ac, ab, and bc unit cell faces, respectively (Figure R1).

The parameters are list in the following table.

	Lattice parameter (nm)			Surface metal density ($10^{15} \text{ atoms cm}^{-2}$)			
	a	b	c	ac	ab	bc	average
Ni_{12}P_5	0.8629	0.8629	0.5036	1.38	0.805	1.38	1.19

Therefore, the L and TOF was calculated as

	L ($\mu\text{mol g}^{-1}$)	TOF (s^{-1})
Ni_{12}P_5	183.0	0.24
10.4 wt% $\text{Ni}_{12}\text{P}_5/\text{SiO}_2$	1210.6	0.22
5.2 wt% $\text{Ni}_{12}\text{P}_5/\text{SiO}_2$	1748.7	0.11
3.1 wt% $\text{Ni}_{12}\text{P}_5/\text{SiO}_2$	1967.3	0.047

In general, both of the above TOF results indicate that the TOF of pristine Ni_{12}P_5 is higher than those of supported versions. We assume one reason is the light harvesting efficiency. In each test the total amount of catalyst (including the Ni_{12}P_5 , and SiO_2 support if available) was controlled as ~ 0.5 mg, thus the higher loading of Ni_{12}P_5 result in a higher light harvesting efficiency, as indicated in the UV-vis-NIR absorption spectra. Consider the photothermal mechanism, a higher light absorption could lead to a higher local temperature, and the local temperature is exponential to the reaction rates.

In response to the reviewer's suggestion on "quantum efficiency", herein we calculated the quantum yield (QY) of these samples. In addition, to exclude the light absorption variations and make a fair comparison, the internal quantum yield was used, which can be defined as:⁴

$$\text{Internal quantum yield} = \frac{\text{produced CO molecules per unit time}}{\text{absorbed photon numbers per unit time}}$$

The absorbed number of photons per unit time, N_{photon} , is estimated from the light intensity dispersion of the Xe lamp (Figure R2) and the UV-vis-NIR absorption spectra: according to:

$$N_{\text{photon}} = \int_{300 \text{ nm}}^{2400 \text{ nm}} \frac{\text{Light intensity} * I\% * A\% * \text{illumination area} * \text{time}}{\text{Average single photon energy} * N_A}$$

Where the light intensity is 2.3 W, illumination area is 1 cm^2 , I% is the percentage of the Xe light intensity at certain wavelength, A% is the light harvesting efficiency at certain wavelength according to the absorption spectra (Fig. 1b in manuscript), time is 3600s, N_A is the Avogadro constant. The average single photon energy (E_{photon}) is figured out using the equation:

$$E_{\text{photon}} = \frac{hc}{\lambda}$$

where h is the Planck constant, c indicates speed of light, and λ is the wavelength.

Thus, we can figure out that total incident photon flux from the lamp is about 5.49 mmol/h, while for the Ni_{12}P_5 , 10.4 wt% $\text{Ni}_{12}\text{P}_5/\text{SiO}_2$, 5.2 wt% $\text{Ni}_{12}\text{P}_5/\text{SiO}_2$, 3.1 wt% $\text{Ni}_{12}\text{P}_5/\text{SiO}_2$ samples they can absorb 92.3%, 73.2%, 55.5%, 47.7% of the photon flux, respectively. The quantum yield results are listed in the following table. From the quantum yield estimation, we can find that lower loading ones have inferior QY values. However, the QY of Ni_{12}P_5 and 10.4 wt% $\text{Ni}_{12}\text{P}_5/\text{SiO}_2$ are quite similar.

Generally, the quantum yield was influenced by both the light harvesting ability and the surface reaction activity. With less light harvesting efficiency (and thereof lower local temperature under light stimulation), the similar photon-to-chemical efficiency on 10.4 wt% $\text{Ni}_{12}\text{P}_5/\text{SiO}_2$ sample when compared with that of the pristine Ni_{12}P_5 sample indicates a possible higher surface catalytic activity of the former.

Although in our test the catalyst weight (including the Ni_{12}P_5 and SiO_2 support) was controlled as ~ 0.5 mg, it is conceivable that the 10.4 wt% $\text{Ni}_{12}\text{P}_5/\text{SiO}_2$ sample would be more competitive when the whole system was scaled up and analyzed, due to its well-balanced internal QY with active Ni_{12}P_5 catalyst usage.

Sample	QY%
Ni_{12}P_5	0.060
10.4 wt% $\text{Ni}_{12}\text{P}_5/\text{SiO}_2$	0.059
5.2 wt% $\text{Ni}_{12}\text{P}_5/\text{SiO}_2$	0.026
3.1 wt% $\text{Ni}_{12}\text{P}_5/\text{SiO}_2$	0.0086

Figure R2. Spectra of the 300 W Xe lamp.

In the revised manuscript, the details of TOF and QY calculations have been added to the Supplementary Information as Note 1 and 2, respectively. Corresponding discussions are also added in the revised manuscript, highlighted in yellow Page 6-8.

References:

1. R. Wojcieszak, S. Monteverdi, M. Mercy, I. Nowak, M. Ziolek, M. M. Bettahar, *Appl. Catal., A* **2004**, 268, 241-253.
2. A. Berenguer, J. A. Bennett, J. Hunns, I. Moreno, J. M. Coronado, A. F. Lee, P. Pizarro, K. Wilson, D. P. Serrano, *Catal. Today* **2018**, 304, 72-79.
3. X. Wang, P. Clark, S. T. Oyama, *J. Catal.* **2002**, 208, 321-331.
4. M. Qureshi, K. Takanahe, *Chem. Mater.* **2017**, 29, 158-167.

4. Figure 5a: This figure should be replaced by Figure R3 from the author rebuttal document since it shows a narrower range of wavenumbers. Also, it would be helpful if the Y axis was expanded so that the IR peaks are larger. The reviewer is not convinced by the author rebuttal that there is no peak broadening as the Ni₁₂P₅ loading is lowered. The authors should calculate and report FWHM for each spectrum to support their argument that there is no peak broadening. Also, there is pronounced asymmetry of the IR peaks that suggests a range of sites for linear bonded CO species. The authors should comment on this point.

Re: Again, grateful for this reviewer's point on FWHM. We have estimated the FWHM of the FTIR peaks, as shown in the following Table R1. With decreased loading the FWHM shows a small change.

We accept the reviewer's suggestion. In the revised manuscript:

(1) The Figure 4a was changed to show an expanded view in which the spectra are plotted within a narrower range of wavenumbers. The old survey spectra are moved to the supplementary information.

(2) Comments on peak broadening and asymmetry were added, yellow highlighted changes in Page 10. In brief, the decreasing particle size would result in more pronounced peak broadening and asymmetry, indicating the likely existence of a range of sites for linearly bonded CO species in the smaller particles. Note we also emphasized that the linearly bonded CO on the single sites of the Ni₁₂P₅ surface would not change much despite size and shape variations.

Table R1. The FWHM estimation of the FTIR peaks of linear bonded Ni-CO species.

Sample	FWHM (nm)
10.4 wt% Ni ₁₂ P ₅ /SiO ₂	13.1
5.2 wt% Ni ₁₂ P ₅ /SiO ₂	13.9
3.1 wt% Ni ₁₂ P ₅ /SiO ₂	17.2

5. Figure 5b-d: The surface crystal structures correctly show that there are two distinct types of Ni sites in Ni₁₂P₅? Might these distinct Ni sites have different interactions with CO₂ and CO? This could explain the peak broadening of the IR spectra. Oyama has proposed that the distribution of tetrahedral Ni(1) and square pyramidal Ni(2) sites at the surface of Ni₂P particles changes with particle size (based on EXAFS data). This could be relevant for Ni₁₂P₅ also. A recently published DFT study (Energy Technology 2019, 7, 1900013; <https://doi.org/10.1002/ente.201900013>) examines the low Miller index surfaces of Ni₁₂P₅ with a focus on determining potential active sites (for HER). The results of this study should be considered and discussed in the context of the proposed atomically precise Ni clusters being the active sites for the photothermal RWGS reaction.

Re: Thanks for the information offered by the reviewer. Following the reviewer's suggestion in the revised manuscript the comments on the asymmetry of the FTIR peak broadening have been added (yellow highlights Page 10). This phenomenon could relate to the two types of Ni site in Ni₁₂P₅ and their different arrangements on different exposed facet. However, overall, the linear bonded Ni-CO is still more preferred than the bridged bonded Ni-CO, due to the ensemble effect of P and the expanded Ni-Ni distance.

In the revised manuscript the following references are also cited:

1. J. Hu, W. Chen, X. Zhao, X. Cao, J. Zhu, Z. Chen, *Energy Technology* **2019**, 7, 1900013.
2. H. Zhao, S. T. Oyama, H.-J. Freund, R. Włodarczyk, M. Sierka, *Appl. Catal., B* **2015**, 164, 204-216

Reviewer #2 (Remarks to the Author):

The authors revised and improved the manuscript taking most comments of the reviewers into account. The great efforts are highly appreciated. After going through all answers and actions taken and carefully reading again the whole manuscript, I conclude that the reliable novelties and the details are still unclear, despite that the contents are well presented in a way that the messages are well highlighted. Therefore, I still do not think the quality of the manuscript meets the standard of the journal under consideration. A few further remarks are given below.

Re: Thanks for the reviewer's suggestions.

- As commented also by the other reviewer, the origin of the catalytic activity is not clear. If it is by photothermal effects, the thermal catalytic activity should be more systematically studied. Probably checking other more standard materials in the reactor

system would improve the reliability of this work.

Re: Thanks for reviewer's comments. In the revised manuscript we have now tested and presented results for the performance of the commercial iron-chrome high temperature reverse water gas shift catalyst, HiFUEL® W210 in Supplementary Table 2. Under the same test conditions, the HiFUEL® W210 delivered a CO production rate of 63.0 mmol g_{cat}⁻¹ h⁻¹, with selectivity of 99.9%. The rate is significantly lower than those of the pristine Ni₁₂P₅ and supported Ni₁₂P₅/SiO₂ samples.

- Higher catalytic activity with illumination is reported, but this can be simply the effect of heating by the energy of the lamp. The "photothermal" effect needs to be carefully studied. Can the catalyst temperature be measured?

Re: We understand the reviewer's concern.

In our tests on a batch reactor, we find under 2.3W of illumination the reactor could be heated up to 60 °C by the lamp, which is much lower than the simulated surface temperature caused by photothermal effect (about 380-400 °C).

As for flow reactor system, the tests are conducted under a pre-determined temperature controlled by using a thermal couple. Therefore, in both the light condition and dark condition the apparent temperature is controlled in the same way.

- The rationale behind the choice of the material should be clearly given. According to Supplementary Table 2, there does not seem a consistency for a choice of the materials in literature. An explanation is expected.

Re: As for photothermal catalysis, firstly, the catalyst should offer a highly catalytic surface toward the targeted chemical reaction. This requires the catalyst can afford a high turnover rate, along with a satisfactory reaction selectivity under thermal activation.

Secondly, since the chemical reaction is driven by the light irradiation, sufficient light absorption across the solar spectra by the catalyst is essential. Therefore, a broad light absorption wavelength range, along with a high extinction coefficient is preferred. In addition, the material should also have a low thermal emissivity, which could prevent the heat loss under irradiation.¹

Other factors such as stability, cost-efficiency, environmental-friendliness, should also be considered.

As for our Ni₁₂P₅ catalyst, its UV-vis-NIR absorption spectra indicate its ability to absorb the full solar spectrum. Moreover, it is a highly active and selective RWGS reaction catalyst. The above advantages enabled its high photothermal performance under simulated solar irradiation. A continuous 100-h test under catalytic working

conditions showcases its high stability. In addition, Ni₁₂P₅ is composed of earth abundant elements, and has appealing cost metrics when compared to the rare and expensive Pt, Pd, In and Cu- based catalysts listed in Supplementary Table 2.

1. Y. Li, J. Hao, H. Song, F. Zhang, X. Bai, X. Meng, H. Zhang, S. Wang, Y. Hu, J. Ye, *Nat. Commun.* **2019**, *10*, 2359

- The precise and defined nature of the active sites are claimed. Even a single site nature and surface structure are presented and discussed, although their relationships with catalytic activity are not established. Elucidating this point is likely necessary to be impactful for the readers of the journal under consideration. I find Figure 1 is still misleading and confusing.

Re: In the revised manuscript we have removed the conceptual Fig.1 and related statements on “atomic precise”. More details are shown in the response to Referee 1’s comment #1. Hopefully this will suffice.

- The chromatograms shown in Supplementary Fig. 5 show that there are issues with gas separation and consequently the quantification.

Re: Thanks for the reviewer’s careful reading. As shown in the GC spectra, the CH₄ peak and CO peak do not overlap with each other. Each peak drops back to the baseline before the next peak emerges. In fact, the peak of CH₄ and CO has an interval of 0.6 min and 0.7 min in the GC spectrum test on the batch reactor system and flow reactor system, respectively, which is sufficient to separate them. In this study the CO production overwhelms the CH₄ production, which results in the huge difference of the peak size of these two components.

Figure R2. The GC traces from catalyst testing in **a**, batch reactor and **b**, flow reactor. The insets are corresponding expanded view.

- The catalytic activity reported in this study is much higher than those presented in literature (Supplementary Table 2). Based on the values given in this work, I quickly calculated the maximum amount of CO produced (i.e. at 100% H₂ conversion) and I

obtained about 200 mmol g_{cat}-1 (no h-1 scaling – it will be much lower in practice by h-1 scaling) as the maximum value (in reality I guess 1 order of magnitude less). In this work about 1000 mmol g_{cat}-1 h-1 is reported (i.e. 28 g CO g_{cat}-1 h-1 which is impossibly high under any condition using any catalyst). I guess the values must be checked carefully

Re: Thanks for the reviewer's careful reading. Here are our calculation details:

Firstly, according to the ideal gas law: PV=nRT, the parameters and molar amount of initial CO₂ (15 psi) and H₂ (3 psi) are listed in the following table

	P	P	V	T	R	n
	psi	Pa	m ³	k	J/(mol·k)	μmol
H ₂	3	20684.27	0.0000118	333.15	8.31451	88.11
CO ₂	15	103421.36	0.0000118	333.15	8.31451	440.58

* 1 psi=6.8947 Pa

Take one run on the best sample 10.4 wt% Ni₁₂P₅-SiO₂ for example. After 0.5 h of illumination, about 28.5 μmol CO was produced (from GC analysis). Consider the catalyst amount of 0.55 mg, and loading amount of 10.4%, the CO production rate was calculated as:

$$\text{Rate} = \text{CO production} / (\text{mass of catalyst} * \text{loading amount}) = 996.5 \text{ mmol g}_{\text{cat}}^{-1} \text{ h}^{-1}$$

Under this circumstance, the conversion of CO₂ and H₂ is 6.4% and 32.3%, respectively. The conversion of CO₂ at this test duration is in accordance with that in Supplementary Fig. 9. It is noteworthy the conversion of H₂ is very high, implying efficient usage of the more valuable H₂ feedstock.

While the above data is just one run, the results shown in Table 1 and Figure 3a are average value from multiple runs.

Reviewer #3 (Remarks to the Author):

I appreciate the efforts by the authors in response to the feedbacks by the reviewers. However, I found that not all responses by the authors are substantiated or reflected clearly in the manuscript.

Re: We thank the reviewer for these helpful comments and acknowledgement of our efforts.

1. I still find it premature in regards to the atomic perfection claim by the authors. Even the authors agreed that defects may be present which led to the formation of CH₄ during catalysis. In addition, faint shoulder at 1950 cm⁻¹ which corresponds to bridging

CO can be observed in Figure R3.

Re: Thanks for reviewer's careful reading. Yes, there is a weak peak, which can be assigned to bridged-bonded Ni-CO, which we have already described in the manuscript. In addition, in the revised manuscript the in-situ DRIFT plots were changed to an expanded view, which show both the linear-bonded Ni-CO and bridged-bonded Ni-CO peaks clearly.

The point regarding the concept of "atomic perfection" was also raised by Referee 1 and responded to in comment #1, please refer to this which will hopefully suffice.

2. The authors mentioned about the drop activity and selectivity temperature > 350 °C. However, no quantitative information was presented in Figure 5 or S11.

Re: We did not report the activity and selectivity drop above 350 °C. In fact, the time-resolved in situ XANES curves under CO₂/H₂ flow conditions indicated that the Ni₁₂P₅ is highly stable at 350 °C (Fig. 2c).

Note in the discussion part we do mention that in a published paper a Ni single atom catalyst was synthesized and used for the CO₂ hydrogenation reaction. In that study the authors mentioned that temperatures above 350 °C could cause serious particle aggregation problems, which is harmful to the reactivity and selectivity. This effect is not found in our Ni₁₂P₅ case. Respectfully, the reviewer may have misunderstood.

3. In response to reviewer 3, the authors stated that catalyst tests were performed at different wavelengths, and suggest the photothermal nature of the catalyst based on the supposed results. However, it was not convincing as these observations were not discussed nor presented in the revised manuscript. I find it odd to exclude such direct observations from the manuscript. The clarification of the catalytic nature of the catalyst may be important with regards to the discrepancy between the batch and flow reactor.

Re: Thanks for the reviewer's comments. In this study we conducted the photothermal catalysis under different light intensities, with the assistance of a set of neutral-density filters. In a typical photochemical process, the chemical reaction is driven by the photo-generated charge carriers. While the number of photo-generated charge carriers scale with the light intensity (incident photon number), the production rate tends to scale linearly with the light intensity. However, in this study the plot of CO production rate versus light intensity (Figure 3b) indicate an Arrhenius-like exponential dependence, implying a dominant photothermal mechanism, where the catalyst absorbs and converts the incident photon into heat to drive surface catalytic reactions.

In another set of experiments in which the photocatalytic tests were conducted at different wavelengths obtained with cut-off filters, we find both visible light and NIR light were able to activate the Ni₁₂P₅ catalyst and promote the RWGS reaction. Moreover,

the plot of CO production rate versus light intensity can also be fitted by an Arrhenius-like exponential function, indicating the CO production rate is predominantly related to the intensity of the light rather than the wavelength, indicating a predominantly photothermal mechanism.

In response to the reviewer's concerns, in the revised manuscript the discussion on the photothermal effect has been expanded (yellow highlighting in Page 8). Moreover, the estimation on quantum yield has also been added to enhance our discussions.

REVIEWER COMMENTS

Reviewer #1 (Remarks to the Author):

The authors have addressed each of this reviewer's questions and comments constructively and the reviewer suggests that the manuscript is ready for acceptance once one remaining point has been addressed.

1. "Single site": This term is used to describe catalytic sites on Ni₁₂P₅ in multiple spots in the manuscript (Abstract – two times, Figure 4 caption, page 10 (line 12), Discussion (1st sentence)). The reviewer recommends that this descriptive term be removed from the manuscript as there is no direct evidence that the Ni₁₂P₅ nanoparticles are single site catalysts. Two reviewers expressed objections about the conclusion of Ni₁₂P₅ being "atom precise" and "single site" catalysts in previous versions of the manuscript. While the "atom precise" description was removed from the most recent revision, the "single site" description remains.

Reviewer #3 (Remarks to the Author):

I find the current iteration of revision to cast further doubts in the authors' presented work. The authors have decided to remove a significant novelty from their work rather than attempting to substantiate it. I do not find the newly chosen "nanocluster" a fitting description of the presented catalysts without clear justifications. With the significant overhanging questions still associated with this revision, I do not find the response to the reviewers to be adequate.

1. Rather than providing further evidence to prove the atomic feature of the catalyst, the authors decided to remove the "atomically precise" feature. I feel that a significant novelty of the work is missing in this particular revision. As commented by all reviewers, elucidation of the "atomically precise" claim is necessary for the work to be impactful for the readers of the journal under consideration.

2. I am not sure if nanoclusters is a fitting description of the presented catalysts based on the size distributions (average sizes are 8 - 13 nm) presented in Supplementary Figure 3.

3. I am in doubt of the quantum efficiency calculation presented with regards to Reviewer 1 comment 3. From the catalytic performance chart presented in Figure 3, Ni₁₂P₅ has the lowest overall activity (155.7 mmol g⁻¹ h⁻¹) while having the highest absorbance (Figure 1b). How did the authors end up with the highest calculated internal quantum yield (based on total CO generated/light adsorbed) for Ni₁₂P₅? As such, I find the discussion on page 7-8 to be highly misleading.

4. Also, the author did not attempt to clarify the confusion brought up by Reviewer 3, comment 2 in the revised manuscript. As the authors did not present any activity at 350 oC, it seems irrelevant to mention this in their discussion. It is not clear whether they are referring to their results or cited work.

5. As there are other catalysts with high selectivity towards CO generation, I am not sure if having a high specific activity alone is sufficient to justify the publication of this work in this journal without a clear mechanistic explanation.

Some additional issues that I found:

6. The authors claimed that "The merit of linearly bonded CO on single Ni sites lies in the relative weaker binding energy (page 10), which would facilitate the desorption of produced CO." What is reference for the comparison to is not clearly explained by the authors.

7. Figure 4d-g are missing in the revised manuscript.

Response to the Reviewer's Comments

Reviewer #1 (Remarks to the Author):

The authors have addressed each of this reviewer's questions and comments constructively and the reviewer suggests that the manuscript is ready for acceptance once one remaining point has been addressed.

Re: We were delighted to get the reviewer's recognition of the acceptability of our work. The reviewer's insightful feedback proved most valuable in enhancing the quality of the work reported in our paper.

1. "Single site": This term is used to describe catalytic sites on Ni₁₂P₅ in multiple spots in the manuscript (Abstract – two times, Figure 4 caption, page 10 (line 12), Discussion (1st sentence)). The reviewer recommends that this descriptive term be removed from the manuscript as there is no direct evidence that the Ni₁₂P₅ nanoparticles are single site catalysts. Two reviewers expressed objections about the conclusion of Ni₁₂P₅ being "atom precise" and "single site" catalysts in previous versions of the manuscript. While the "atom precise" description was removed from the most recent revision, the "single site" description remains.

Re: Appreciate the reviewer's valuable suggestion. In the revised manuscript the descriptor "single site reaction route" has been removed or replaced by the term "linearly bonded nickel-carbonyl-dominated reaction route"

Reviewer #3 (Remarks to the Author):

I find the current iteration of revision to cast further doubts in the authors' presented work. The authors have decided to remove a significant novelty from their work rather than attempting to substantiate it. I do not find the newly chosen "nanocluster" a fitting description of the presented catalysts without clear justifications. With the significant overhanging questions still associated with this revision, I do not find the response to the reviewers to be adequate.

Re: Thanks for reviewer's comments. We hope the response to the comments voiced below could help the reviewer catch the points and appreciate the novelty of this work.

1. Rather than providing further evidence to prove the atomic feature of the catalyst, the authors decided to remove the "atomically precise" feature. I feel that a significant novelty of the work is missing in this particular revision. As commented by all reviewers, elucidation of the "atomically precise" claim is necessary for the work to be impactful for the readers of the journal under

consideration.

Re: We understand the reviewer's concern.

The “atomically precise” concept was put forward based on our experimental observations and crystal structure of Ni_{12}P_5 . In detail, firstly, the XPS and XANES indicated that the Ni in Ni_{12}P_5 was close to its metallic state. However, the Ni, a well-documented CO_2 methanation catalyst, when in the form of Ni_{12}P_5 it turned out to be an impressive CO producer with near unity selectivity. A similar phenomenon has only been observed in the case of a Ni single atom catalyst or a Ni centered homogeneous catalyst. Thus, together with the crystal structure of Ni_{12}P_5 , we thereby rationalized that the active site should be the well-separated periodic arrays of few atom Ni clusters in Ni_{12}P_5 .

Furthermore, based on the in-situ DRIFT spectra of various samples, and results of catalytic tests under different conditions (e.g., particle size, temperature, initial CO_2/H_2 ratio), it was found that the reaction pathway and the reaction selectivity were well sustained on Ni_{12}P_5 catalyst with various size and shape. These observations thus provided credence that the surface CO_2 hydrogenation reaction on Ni_{12}P_5 could be regulated by so-called “atomic perfect Ni nanoclusters”.

Recall, as the reviewers mentioned previously, there always exists surface imperfections, and exposure of multiple facet due to the different crystal sizes and morphologies. Clearly, in this study the Ni_{12}P_5 nanoparticle was not an ideal model to meet the reviewer's expectations, thus after careful consideration and due cognizance to the reviewer's request, we removed the “atomically precise nanocluster” descriptor, and use “nanocluster” instead.

Nevertheless, we believe the efforts we made in this study to unlock the light-driven CO_2 hydrogenation ability of metal phosphides, has still offered impactful and valuable insights for researchers interested in expanding upon and enhancing the advance reported in our first paper on the subject – note that this point was also addressed in Reviewer 1's comments in the first round peer-review.

On the one hand, in this paper our results demonstrated that the photothermal catalytic CO_2 hydrogenation activity of the archetype Ni_{12}P_5 is quite impressive when compared to other reported materials, especially those containing the noble metal. Based on our multi-analytical studies, we think the highly active surface together with the excellent light harvesting behavior should be responsible for the striking performance metrics of Ni_{12}P_5 .

As mentioned, we also highlighted the near-unit reaction selectivity and offered a rational explanation on the reaction pathway consistent with the experimental results and the reported crystal structure.

Combined with the demonstrated extension of the Ni_{12}P_5 idea to the Co_2P analogue, we believe this work could provide an interesting steppingstone for (photo)thermal catalytic carbon dioxide conversion over an extensive family of metal phosphide materials.

2. I am not sure if nanoclusters is a fitting description of the presented catalysts based on the size distributions (average sizes are 8 - 13 nm) presented in Supplementary Figure 3.

Re: The reviewer is right that a size of few nanometers for a material is not in the scope of nanoclusters. Note we do not define the prepared Ni₁₂P₅ and Ni₁₂P₅-SiO₂ material as a “nanocluster”, rather we always use the term “nanoparticle” to describe them.

However, as presented in the introduction to the paper, at the scale of the unit cell we write: *“Metal phosphides form a class of solids that provide the envisioned atomic and crystalline perfection with a structure based upon highly dispersed metal nanoclusters chemically integrated in a P lattice.”*

When it comes to the surface, which has great influence on the reaction pathway, one can see that the surface of Ni₁₂P₅ is composed of Ni clusters (2-4 atoms) separated by P atoms. This is seen in the significantly decreased coordination number of Ni in Ni₁₂P₅ when compared to the bulk metallic Ni (already shown in Supplementary Table 1).

Considering the materials structure is highly stable during the catalytic process (revealed by in-situ XANES) taken in conjunction with the catalytic results, which implies the reaction center is comprised of few-atom nanoclusters (please see Discussions part), therefore in the statements relating to the reaction pathway we now use the term “nanocluster”.

Hopefully this explanation clarifies what we mean in our discussion related to the scale of the synthesized nanoparticles and the few-atom nanoclusters embedded within.

3. I am in doubt of the quantum efficiency calculation presented with regards to Reviewer 1 comment 3. From the catalytic performance chart presented in Figure 3, Ni₁₂P₅ has the lowest overall activity (155.7 mmol g⁻¹ h⁻¹) while having the highest absorbance (Figure 1b). How did the authors end up with the highest calculated internal quantum yield (based on total CO generated/light adsorbed) for Ni₁₂P₅? As such, I find the discussion on page 7-8 to be highly misleading.

Re: Here are our explanations:

As shown in Supplementary information, the IQY is calculated according to the following formula:

$$\text{Internal quantum yield}_{\text{CO}} = \frac{\text{produced CO molecules per unit time}}{\text{absorbed photon numbers per unit time}}$$

In the photocatalytic test, the catalyst, Ni₁₂P₅ nanoparticles or SiO₂-supported Ni₁₂P₅ nanoparticles, are loaded on a silica paper substrate with an area of 1 cm². The mass of the catalyst (include the Ni₁₂P₅ and SiO₂ if applicable) is controlled at about 0.5 mg.

When we calculate the CO production **rate** (**unit:** mmol g_{cat}⁻¹ h⁻¹), where the g_{cat} represents the mass of active Ni₁₂P₅ where the mass of inert SiO₂ support was excluded. As shown in the manuscript, the production rate over Ni₁₂P₅-SiO₂ is higher than the unsupported Ni₁₂P₅.

However, because the more active Ni₁₂P₅ sample was loaded on the silica paper in the case of the unsupported Ni₁₂P₅ sample, the produced CO molecules per unit time (**unit**: mmol h⁻¹) is higher than those produced for the SiO₂ supported sample in our test.

To amplify, the average CO production for Ni₁₂P₅ is 70.2 μmol h⁻¹, while for 10.4 wt% Ni₁₂P₅/SiO₂ sample it is 55 μmol h⁻¹. Considering these two samples absorb 5.07 mmol and 4.02 mmol of photons (as presented in Supplementary Note 2), respectively, their corresponding IQY is thereby calculated as 0.060% and 0.059%.

As only about one tenth of the active Ni₁₂P₅ has been utilized in the 10.4 wt% Ni₁₂P₅/SiO₂ sample when compared to pristine Ni₁₂P₅, while their IQY is very close, the former would be more competitive overall, especially if the IQY could be improved and the system scaled-up. Future advances in this respect will require innovative engineering of the photocatalyst and photoreactor to optimize the capture and photocatalytic utilization of the light

4. Also, the author did not attempt to clarify the confusion brought up by Reviewer 3, comment 2 in the revised manuscript. As the authors did not present any activity at 350 °C, it seems irrelevant to mention this in their discussion. It is not clear whether they are referring to their results or cited work.

Re: The discussion related to the “350 °C” as mentioned by the reviewer refers to the findings of a published paper.

To allay the reviewer’s concerns, in the revised manuscript the confusion has been corrected by removing the following misleading statement from the paper.

“However, at high temperature (>350 °C) Ni single atoms would diffuse, nucleate and grow into Ni clusters of about 10 nm, resulting in a drop of activity and selectivity.”

5. As there are other catalysts with high selectivity towards CO generation, I am not sure if having a high specific activity alone is sufficient to justify the publication of this work in this journal without a clear mechanistic explanation.

Re: Firstly, as the reviewer agreed, the pre-eminent activity and selectivity of Ni₁₂P₅ catalyst in the CO₂ hydrogenation reaction process under simulated solar irradiation is one of the highlights of this study. Such impressive catalytic performance appears to be well sustained during a 100-h time-on-stream test in our laboratory. Additionally, Ni₁₂P₅ is composed of earth abundant elements, and has appealing cost metrics when compared to the rare and expensive Pt, Pd, In and Cu- based catalysts listed in Supplementary Table 2. In short, the above advantages endow Ni₁₂P₅ with great potential in photothermal catalytic CO₂ conversion.

To shed more light on the reaction pathway and mechanism, the DRIFT study was adopted. A unique linearly-bonded-CO intermediate dominated reaction route is observed, implying a single-site reaction mechanism, which should be responsible for the near unity selectivity and

the counterintuitive observation that the catalyst performance is essentially independent of nanoparticle size and reaction conditions.

The extension of the ideas presented for the archetype Ni₁₂P₅ to the Co₂P analogue implies that metal phosphide, a large family of materials which has already gain significant recognition in electrochemical catalysis could enjoy similar success as a universal platform for gas-phase heterogeneous CO₂ catalysis. We believe these findings will arouse interest for researchers working on CO₂ thermal catalysis, photocatalysis and photothermal catalysis.

Supplementary Table 2. Comparison of CO₂ conversion rates for some CO₂ hydrogenation catalysts under light irradiation without external heating.

Catalyst	Light source	Feed composition	CO ₂ conversion rate	Selectivity	
				CO	CH ₄
Cu ₂ O ¹	300 W Xe light (full spectrum, 40 suns)	CO ₂ /H ₂ =83/17	70.3 mmol g ⁻¹ h ⁻¹	~100	
In ₂ O _{3-x} ²	300 W Xe light (~20 suns)	CO ₂ /H ₂ =50/50	238.8 mmol g ⁻¹ h ⁻¹	~100	
Pt/NaTaO ₃ ³	300 W UV-enhanced lamp	CO ₂ /H ₂ =50/50 Xe	140.5 μmol g ⁻¹ h ⁻¹	99	1
Pd@Nb ₂ O ₅ ⁴	300 W Xe lamp	CO ₂ /H ₂ =50/50	1.8 m mol g ⁻¹ h ⁻¹	100	
Cu/Pd/HyWO _{3-x} ⁵	300 W Xe lamp (1 W cm ⁻²)	CO ₂ /H ₂ =50/50	40.8 μmol g ⁻¹ h ⁻¹	100	
Fe@C ⁶	300 W Xe lamp	CO ₂ /H ₂ =50/50	26.1 mmol g ⁻¹ h ⁻¹	100	
FeO-CeO ₂ ⁷	300 W Xe lamp (2.2 W cm ⁻²)	CO ₂ /H ₂ /Ar=15/60/25	20 mmol g ⁻¹ h ⁻¹	97-99.9 %	
SA Ni/Y ₂ O ₃ ⁸	ambient daylight sunlight (from 0.52 to 0.7 kW m ⁻²)	CO ₂ /H ₂ /N ₂ =2.5/10/87.5	7.5 L m ⁻² h ⁻¹		100
Ni/SiO ₂ -Al ₂ O ₃ ⁹	solar simulator	CO ₂ /H ₂ /N ₂ =15/70/15	14.4 mmol g ⁻¹ h ⁻¹	2.8	97.2
NiO ⁹	solar simulator	CO ₂ /H ₂ /N ₂ =15/70/15	13.3 mmol g ⁻¹ h ⁻¹		100
iron-chrome based catalyst [†] (this work)	300 W Xe lamp (2.3 W cm ⁻²)	CO ₂ /H ₂ =83/17	63.0 mmol g ⁻¹ h ⁻¹	99.9	0.1
Ni ₁₂ P ₅ (this work)	300 W Xe lamp (2.3 W cm ⁻²)	CO ₂ /H ₂ =83/17	155.7 mmol g ⁻¹ h ⁻¹	99.5	0.5

Ni ₁₂ P ₅ /SiO ₂ (this work)	300 W Xe lamp (2.3 W cm ⁻²)	CO ₂ /H ₂ =83/17	960.3 mmol g ⁻¹ h ⁻¹	99.7	0.3
Co ₂ P (this work)	300 W Xe lamp (2.3 W cm ⁻²)	CO ₂ /H ₂ =83/17	15.7 mmol g ⁻¹ h ⁻¹	99.2	0.8
Co ₂ P/SiO ₂ (this work)	300 W Xe lamp (2.3 W cm ⁻²)	CO ₂ /H ₂ =83/17	227.7 mmol g ⁻¹ h ⁻¹	99.5	0.5

† Commercial iron-chrome based high temperature gas shift catalyst, HiFUEL® W210. The catalyst was milled before using.

Some additional issues that I found:

6. The authors claimed that “The merit of linearly bonded CO on single Ni sites lies in the relative weaker binding energy (page 10), which would facilitate the desorption of produced CO.” What is reference for the comparison to is not clearly explained by the authors.

Re: Thanks for the reviewer’s reminder. The weaker binding energy of linearly bonded CO is compared to the bridge bonded ones. In the latter case the CO molecule is multi-coordinated on Ni with more sigma and pi bonding, which make it more stable and difficult to desorb. In the revised manuscript we have modified the relevant sentence in which the words highlighted in yellow have been added:

“The merit of linearly bonded CO on Ni sites lies in the relative weaker binding energy **when compared to those of bridge bonded species**, and would facilitate the desorption of produced CO.”

7. Figure 4d-g are missing in the revised manuscript.

Re: Thanks for the reviewer’s reminder, Fig 4d-g is now corrected to read Fig. 5. In the last version we forget to delete related caption, while in the current version we have revised it, so all is now fine.

On a final note we wish to sincerely thank the reviewers and the editor for their exceptionally valuable peer review of this paper, which thanks to their guidance has allowed us to produce a much higher quality contribution and hopefully a great home in Nature Communications.

REVIEWERS' COMMENTS:

Reviewer #3 (Remarks to the Author):

I recommend the publication of this work in its current state. I appreciate the efforts of the authors to clarify the confusions raised in the previous iteration. Great job!

Response to the Reviewer's Comments

Reviewer #3 (Remarks to the Author):

I recommend the publication of this work in its current state. I appreciate the efforts of the authors to clarify the confusions raised in the previous iteration. Great job!

Re: We gratefully appreciate the reviewer's valuable comments and the recognition of our efforts.